**TOOLS**

# Small GTPase ActIvitY ANalyzing (SAIYAN) system: A method to detect GTPase activation in living cells

Miharu Maeda[1], Masashi Arakawa[1], Yukie Komatsu[1], and Kota Saito[1]

**Small GTPases are essential in various cellular signaling pathways, and detecting their activation within living cells is crucial for understanding cellular processes. The current methods for detecting GTPase activation using fluorescent proteins rely on the interaction between the GTPase and its effector. Consequently, these methods are not applicable to factors, such as Sar1, where the effector also functions as a GTPase-activating protein. Here, we present a novel method, the Small GTPase ActIvitY ANalyzing (SAIYAN) system, for detecting the activation of endogenous small GTPases via fluorescent signals utilizing a split mNeonGreen system. We demonstrated Sar1 activation at the endoplasmic reticulum (ER) exit site and successfully detected its activation state in various cellular conditions. Utilizing the SAIYAN system in collagen-secreting cells, we discovered activated Sar1 localized both at the ER exit sites and ER–Golgi intermediate compartment (ERGIC) regions. Additionally, impaired collagen secretion confined the activated Sar1 at the ER exit sites, implying the importance of Sar1 activation through the ERGIC in collagen secretion.**

## Introduction

Small GTPases, cycling between inactive guanosine diphosphate (GDP) and active guanosine triphosphate (GTP) forms, serve as molecular switches in various cellular processes, including cell proliferation, growth, differentiation, motility, and membrane trafficking (Cherfils, 2023; Cherfils and Zeghouf, 2013; Takai et al., 2001). The activation of these small GTPases is catalyzed by guanine nucleotide exchange factors (GEFs), which destabilize the binding of GDP to GTPase, thereby facilitating its transition to the free form capable of recruiting cytoplasmic GTP. Upon binding to GTP, the protein interacts with effectors to execute cellular functions. Generally, the intrinsic activity of small GTPases is relatively slow, and their inactivation is facilitated by GTPase-activating proteins (GAPs) (Saito et al., 2017).

The conventional approach to assessing GTPase activity in vitro relies on the use of radioisotopes (Goody, 2016; Northup et al., 1982). Methods, such as differential tryptophan fluorescence measurements of small G proteins in their GDP- and GTP-bound states, can be used as potential improvements to these assays for GTPases containing tryptophan residues (excluding the Ras family) (Antonny et al., 2001; Goody, 2016). Another approach involves the use of a fluorescent nucleotide, such as mant-GTP (Neal et al., 1990).

In addition to the in vitro assays, the measurement of GTPase activity within cells is essential, as it directly reflects the broad cellular responses mediated by GTPases with effector proteins.

There are currently three primary methods for measuring the activity of small GTPases within cells. Initially, $^{32}$Pi labeling of cells was used to directly measure the ratio of bound nucleotides to small GTPases (Gibbs et al., 1987; Satoh et al., 1988). Pulse-labeled cells were extracted, followed by immunoprecipitation of small GTPases, and the bound nucleotides were resolved using thin-layer chromatography to determine the proportion of GDP to GTP. Although highly sensitive, this assay requires radioisotope labeling, is time-consuming, and lacks the ability to monitor changes over time.

Alternatively, the pull-down assay offers a method for measuring activity (de Rooij and Bos, 1997; Taylor and Shalloway, 1996). In this method, cells are extracted, activated small GTPases are captured by effector proteins, and the number of bound GTPases is quantified using western blotting. Despite being relatively straightforward, this assay still requires cell extraction, and GTPase hydrolysis needs to be taken into account during experiments.

The third assay involves the timely detection of activated GTPases within living cells using fluorescent proteins and an effector domain that binds to the GTPase. One method, using a relocation biosensor, involves fusing fluorescent proteins to the effector domain of the GTPase to observe its activation state (Pertz and Hahn, 2004). This method has undergone various refinements to date, as it directly binds to activated GTPases, thereby revealing the site of activation. However, drawbacks

[1]Department of Biological Informatics and Experimental Therapeutics, Graduate School of Medicine, Akita University, Akita, Japan.

Correspondence to Kota Saito: ksaito@med.akita-u.ac.jp.

include the potential for unbound sensors to contribute to background signal, the possibility of recognizing non-specific activity if the effector domain's specificity to the GTPase is not high, and the potential to inhibit endogenous signal transduction pathways (Mahlandt et al., 2021).

Another approach involves measuring the interaction between the GTPase and its effector using a fluorescence resonance energy transfer (FRET)-based biosensor (Aoki and Matsuda, 2009) to assess GTPase activity within cells in real-time. The Raichu, Ras superfamily, and interacting protein chimeric unit represent the first intramolecular biosensors designed to detect FRET between the yellow and cyan fluorescent proteins (YFP and CFP) upon Ras activation and its interaction with the effector Raf (Mochizuki et al., 2001). While this method has been successfully utilized for various GTPases, it has a drawback in that the activation state of this sensor is not necessarily indicative of the activation site of the endogenous GTPase. This is because the activation state of the sensor can be controlled by the activation states and respective localizations of GEFs and GAPs rather than directly reflecting the activation site of the endogenous GTPase. Moreover, the overexpression of FRET sensors in cells restricts the monitoring of endogenous protein activity to specific cellular locations. Furthermore, these methods, which rely on the binding between the GTPase and the effector domain, are not suitable for assessing the activity of small GTPases, such as Sar1, where the effector, Sec23, functions not only as an effector but also as a GAP.

Split fluorescent systems offer versatile tools for a wide array of applications, including systematic endogenous tagging for visualizing endogenous protein localization and detecting membrane contact sites (Cabantous et al., 2005; Kakimoto et al., 2018; Magliery et al., 2005; Romei and Boxer, 2019). In this study, we used this system to measure the activation of endogenous small GTPases in living cells. Our approach allows the detection of Sar1 activation under various physiological conditions and in various locations. Furthermore, this novel system has the potential for application across a broad spectrum of different GTPases, and its effectiveness is discussed in subsequent sections.

## Results

### Validation of SAIYAN technology

Sar1 is a small GTPase with an amphipathic N-terminus belonging to the Arf-family (Van der Verren and Zanetti, 2023). Sar1, in its GDP-bound form, is either cytosolic or weakly associated with the endoplasmic reticulum (ER) membrane. Upon activation by the GEF, Sec12, Sar1 inserts its N-terminal amphipathic helix fully into the membrane, resulting in a strong association with the ER membrane to facilitate the recruitment of Sec23/Sec24 (Barlowe and Schekman, 1993; Bi et al., 2002; Huang et al., 2001; Paul et al., 2023). Sec23 acts both as an effector by interacting with activated Sar1 and as a GAP for Sar1 (Yoshihisa et al., 1993). This dual role poses challenges for the development of conventional FRET-based sensors to monitor Sar1 activity.

To detect Sar1 activation within living cells, we initially established stable cell lines expressing 10 of the 11 β-strands of

mNeonGreen2 ($mNG_{1-10}$) tethered to the ER membrane in HeLa cells, which were obtained by fusion with the membrane-spanning region of TANGO1S and a human influenza hemagglutinin (HA)-tag (Fig. 1 A). This cell line was named HA-$mNG_{1-10}$. Following doxycycline induction, we verified that the constructs labeled with HA were distributed throughout the ER, exhibiting a characteristic reticular staining pattern that colocalized with protein disulfide-isomerase (PDI) (Fig. S1 A, upper panel). However, this construct did not produce any mNeonGreen2 (mNG) signal (Fig. S1 A, upper panel). Next, we transiently expressed Sar1A constructs containing a FLAG-tag, followed by a glycine linker fused to the 11th strand of mNG (Sar1A-FLAG-$mNG_{11}$) in HA-$mNG_{1-10}$ cells (Fig. 1 A). No mNG signal was detected in the absence of HA-$mNG_{1-10}$ induction (Fig. S1 B, upper panel). However, upon induction of HA-$mNG_{1-10}$, mNG signals became apparent (Fig. 1 B, upper panel; Fig. S1 A, bottom panel; Fig. S1 B, bottom panel). These mNG signals exhibited a pattern distinct from those of HA or PDI, appearing as punctate structures in the ER membrane network (Fig. S1 A, bottom panel). Costaining with Sec16, a well-established ER exit site marker, revealed mNG localization at these sites (Fig. 1 B, upper panel) (Watson et al., 2006). Notably, FLAG staining, representing the entire Sar1A protein irrespective of its nucleotide status, was distributed in both the cytoplasm and ER exit sites, maintaining its localization regardless of the expression of HA-$mNG_{1-10}$ (Fig. S1 B). This observation indicated that Sar1A-FLAG-$mNG_{11}$ was not artificially recruited to the ER membrane by $mNG_{1-10}$ to form complete mNG proteins. Therefore, we inferred that the mNG signals corresponded to membrane-associated Sar1A and proceeded to further validate this methodology by examining the activation of various Sar1A mutants.

The ΔN mutant, which fails to insert into the ER membrane, did not generate any mNG signals despite expressing Sar1A at a level comparable with that of the wild-type (WT) Sar1A, as indicated by FLAG staining (Fig. 1 B) (Lee et al., 2005). Conversely, the T39N mutant, deficient in GTP binding, exhibited weak but firm signals throughout the ER (Fig. 1 B) (Kuge et al., 1994). This observation is consistent with reports that the GDP-bound form of Sar1 binds to membranes more weakly than the GTP-bound form of the protein (Paul et al., 2023). In contrast, the Sar1A H79G mutant, a GTPase-deficient variant of Sar1A, generated mNG signals comparable with Sar1A WT that colocalized with Sec16 (Fig. 1 B) (Aridor et al., 1995). The quantification of the mNG signal in these results is shown in Fig. 1 C. Compared with the signal from Sar1A WT, there was almost no signal from ΔN, while T39N showed a signal intensity about half that of Sar1A WT. In contrast, H79G showed an intensity almost comparable to Sar1A WT. Considering the biochemical knowledge about the nucleotide forms of Sar1 mutants and the localization of Sec12 at the ER exit site, we believe that the mNG signal at the ER exit site observed with WT and H79G recognizes the GTP-bound form of Sar1 (Saito et al., 2014; Van der Verren and Zanetti, 2023). Based on these findings, we can reasonably infer that, through this approach, we can quantitatively assess the levels of Sar1A bound to membranes. Furthermore, it is plausible to anticipate that we may gather insights into the activation state and site of Sar1A. Thus, we named this technology the Small GTPase ActIvitY

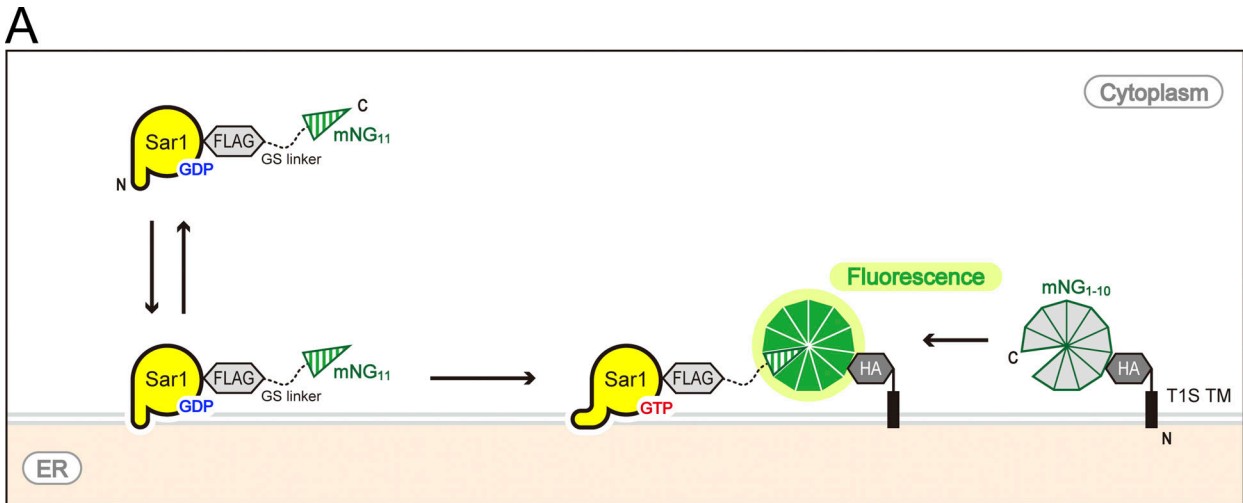

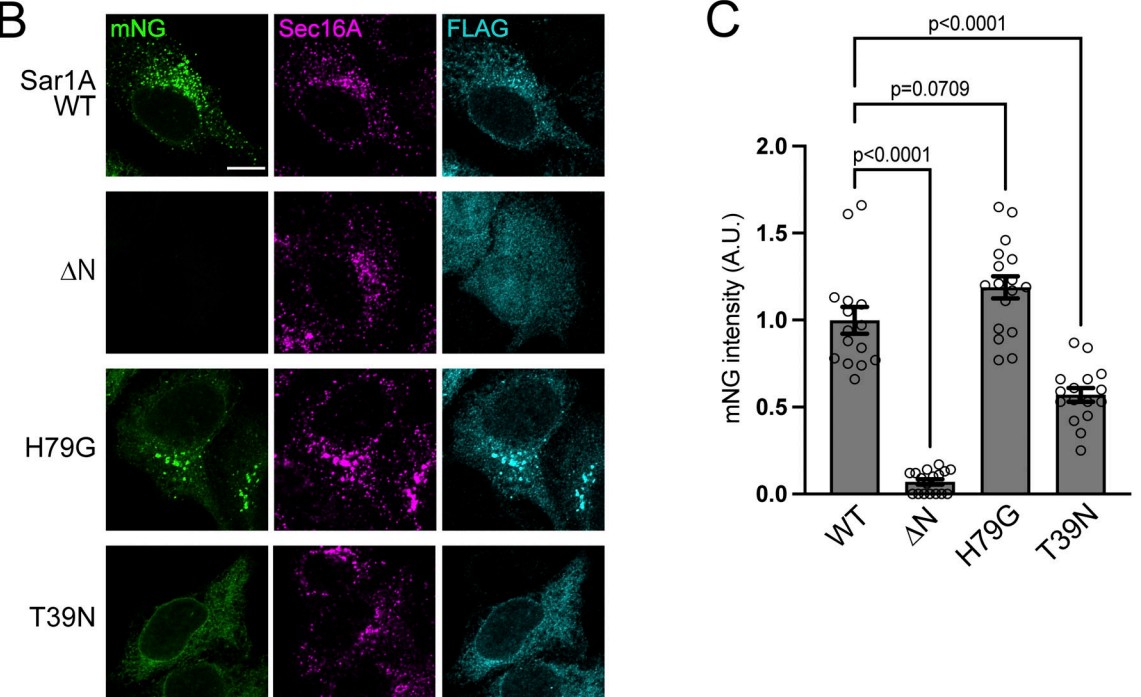

**Figure 1. Sar1 membrane association is efficiently detected using the Split mNeonGreen system. (A)** Schematic representation of the SAIYAN system. The membrane-spanning regions of TANGO1S and HA-tag fused to 10 of the 11 strands of mNG (mNG$_{1-10}$) were expressed in cells. In addition, Sar1A constructs with a FLAG-tag and a glycine linker fused to the 11th strand of mNG (mNG$_{11}$) were also expressed. Upon Sar1A activation, mNG$_{1-10}$ and mNG$_{11}$ combined to form the complete mNG proteins, inducing mNG signals. **(B)** HA-mNG$_{1-10}$ cells transfected with the indicated Sar1A constructs were fixed and stained with anti-Sec16-C and anti-FLAG antibodies. Scale bar = 10 μm. **(C)** Quantification of mNG intensity from B (arbitrary units [A.U.]). Error bars represent the means ± SEM. Each data point represents the mNG intensity of the analyzed cells.

ANalyzing (SAIYAN) system and established cell lines capable of detecting endogenous Sar1A activation. Endogenous Sar1A was tagged with mNG$_{11}$ using CRISPR/Cas9-mediated knock-in tagging of HA-mNG$_{1-10}$ cells (Fig. 2 A). These cell lines were validated by sequencing the Sar1A genomic locus and were designated as Sar1A/SAIYAN (HeLa) cells. Consistent with the transient expression, mNG signals were observed only upon doxycycline induction and were exclusively localized at the ER exit sites (Fig. 2, B–D).

Suppressing Sar1A expression with siRNAs in Sar1A/SAIYAN (HeLa) cells resulted in diminished mNG fluorescence (Fig. 2 D). Quantification of the mNG signals is presented in Fig. 2 E. Western blotting of cell lysates confirmed the reduction in bands corresponding to Sar1A and FLAG, further validating the production of Sar1A/SAIYAN (HeLa) cells (Fig. 2 F). These results indicate that not all Sar1A genes were tagged with FLAG.

We then examined the localization of Sar1A to the membrane in the presence or absence of doxycycline by biochemical

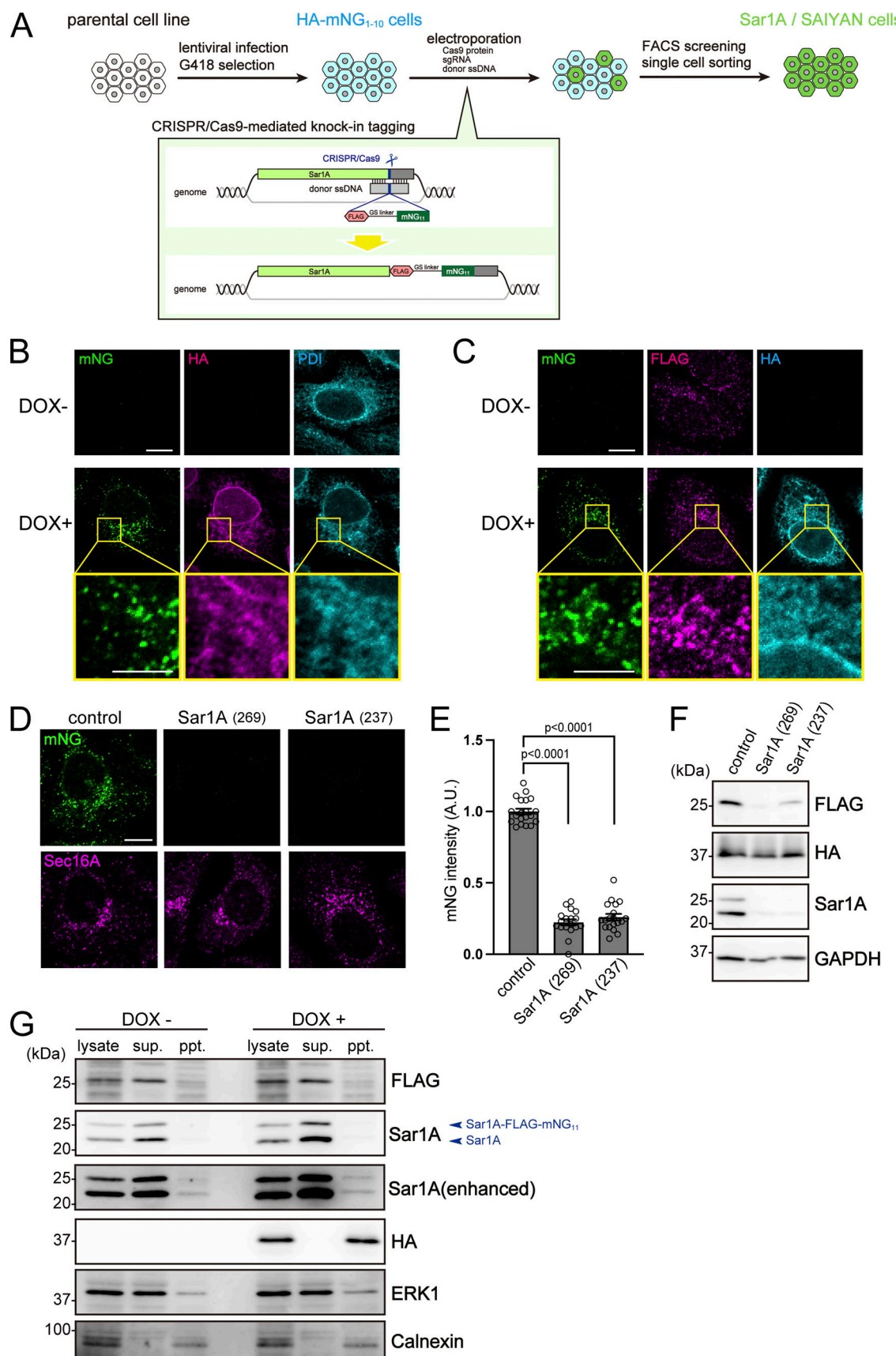

Figure 2.  **Production of Sar1A/SAIYAN cells. (A)** Doxycycline-inducible stable cell lines expressing the membrane-spanning regions of TANGO1S and HA-tag fused to 10 of the 11 strands of mNG were established using a lentiviral system and G418 selection (HA-mNG$_{1-10}$ cells). Stable cells were subsequently

electroporated with Cas9 protein, sgRNA, and ssDNA to facilitate the knock-in of FLAG-mNG$_{11}$ into the Sar1A locus of the genome. Cells were treated with doxycycline for 24 h and further sorted via FACS to isolate single cells exhibiting mNG signals into 96-well plates. The expanded cell population was then collected and subjected to genomic sequencing. Positive clones were identified and selected for further analysis (Sar1A/SAIYAN cells). **(B)** Sar1A/SAIYAN (HeLa) cells, either treated or non-treated with doxycycline, were fixed and stained with anti-HA and anti-PDI antibodies. Boxed areas in the middle panels are shown at high magnification in the bottom panels. Scale bars: 10 µm (main), 5 µm (magnification). **(C)** Sar1A/SAIYAN (HeLa) cells, either treated or non-treated with doxycycline, were fixed and stained with anti-HA and anti-FLAG antibodies. Boxed areas in the middle panels are shown at high magnification in the bottom panels. Scale bar: 10 µm (main), 5 µm (magnification). **(D)** Sar1A/SAIYAN (HeLa) cells transfected with the indicated siRNAs were fixed and stained with an anti-Sec16-C antibody. Scale bar = 10 µm. **(E)** Quantification of mNG intensity from D (arbitrary units [A.U.]). Error bars represent the means ± SEM. Each data point represents the mNG intensity of the analyzed cells. **(F)** Sar1A/SAIYAN (HeLa) cells transfected with the indicated siRNAs were lysed and subjected to SDS-PAGE, followed by western blotting with anti-FLAG, anti-HA, anti-Sar1A, and anti-GAPDH antibodies. **(G)** Sar1A/SAIYAN (HeLa) cells, either treated or non-treated with doxycycline, were fractionated via centrifugation. The lysates, the supernatants, and the pellets were subjected to SDS-PAGE, followed by western blotting with anti-FLAG, Sar1A, HA, ERK1, and calnexin antibodies. Source data are available for this figure: SourceData F2.

fractionation in Sar1A/SAIYAN (HeLa) cells. Both intact Sar1A and Sar1A tagged with FLAG-mNG$_{11}$ were found to be predominantly present in the cytoplasm, with no significant change in their ratio with or without the induction of HA-mNG$_{1-10}$ (Fig. 2 G). This strongly suggests that Sar1A tagged with FLAG-mNG$_{11}$ is not artificially localized to the membrane for the formation of complete mNG proteins.

Moreover, no significant difference in the growth of Sar1A/SAIYAN (HeLa) cells was observed in comparison to that of the parental HeLa cells, both with and without doxycycline induction (Fig. S2 A). Additionally, GM130 staining revealed a typical Golgi structure, while the RUSH assay showed no differences in the trafficking of mannosidase II from the ER to the Golgi with or without doxycycline induction, further confirming that the constructs did not affect secretory pathways (Fig. S2, B and C).

### Sar1 activation was mediated by cTAGE5/Sec12 at ER exit sites in living cells

To investigate whether the SAIYAN system can verify the state of activated Sar1 following the knockdown of factors influencing Sar1 nucleotide form, the effects of Sec12 depletion were examined. Consistent with our expectations, Sec12 knockdown using siRNAs resulted in diminished mNG signals (Fig. 3, A and B; and Fig. S3 A), verifying the role of Sec12 in mediating Sar1 activation at ER exit sites (Barlowe and Schekman, 1993). Compared with the minimal signal observed with Sar1A knockdown (Fig. 2 E), the mNG signal from Sec12 knockdown was approximately half the intensity of the signal observed in WT cells and was similar to the mNG signal observed in cells expressing T39N (Fig. 1 C and Fig. 3 B). These findings suggest that in cells with Sec12 knockdown, the detected mNG signal likely represents GDP-bound Sar1. Thus, while the SAIYAN system cannot quantitatively measure the nucleotide form of GTPases within cells, it is capable of detecting changes in nucleotide forms. This capability is based on the fact that GTP-bound GTPases have a stronger affinity for membrane binding than their GDP-bound counterparts.

Our previous investigations demonstrated that the interaction between Sec12 and cTAGE5 is crucial for the appropriate localization of Sec12 to ER exit sites (Saito et al., 2014) and the depletion of cTAGE5 disperses Sec12 throughout the ER (Tanabe et al., 2016). Therefore, the effect of cTAGE5 depletion on Sar1 was examined. The depletion of cTAGE5 decreased mNG signals

at the ER exit sites, suggesting that activated Sar1 was decreased at ER exit sites (Fig. 3, C and D), while the expression level of Sec12 remained unchanged (Fig. S3 B). These findings indicate that the accurate positioning of Sec12 at the ER exit sites is essential for the effective activation of Sar1 at these sites.

### ER exit site organization was crucial for proper Sar1 activation in living cells

We investigated the effects of perturbing the organization of the ER exit sites on Sar1. Our previous study demonstrated the interaction between TANGO1 and Sec16 and its significance in the ER exit-site organization (Maeda et al., 2017; Saito and Maeda, 2019). Consequently, we depleted TANGO1 and Sec16 and examined their effects on Sar1. The knockdown of both TANGO1 and Sec16 resulted in reduced mNG signaling, indicating the essential role of TANGO1 and Sec16 in facilitating Sar1 activation (Fig. 3, E–H; and Fig. S3, C and D).

### Sec23 acted as a stabilizer of activated Sar1 rather than an inactivator in living cells

Our findings suggest that the absence of crucial ER exit-site components resulted in Sar1 inactivation. To assess whether the SAIYAN system could also detect an increase in activated Sar1, Sec23A was depleted. Given that Sec23 promotes GTP hydrolysis of Sar1 in vitro, its depletion should theoretically enhance the GTP-bound form of Sar1. However, mNG signals were severely diminished in Sec23A-depleted cells (Fig. 4, A and B; and Fig. S3 E). Subsequently, we knocked down Sec31A, which is known to enhance Sec23 GAP activity toward Sar1 by ~10-fold in vitro (Antonny et al., 2001; Bi et al., 2007). As anticipated, Sec31A depletion increased mNG signals, suggesting that the amount of activated Sar1 increased with Sec31A depletion (Fig. 4, C and D; and Fig. S3 F). These results indicate that the SAIYAN system can detect the increase in activated small GTPases.

We then investigated whether the introduction of Sec13/Sec31 could reduce Sar1 activation in the SAIYAN system. We transiently transfected Sec13/Sec31A into Sar1A/SAIYAN (HeLa) cells and detected the mNG signals. The mNG signal was significantly reduced in cells expressing Sec13/Sec31A compared with that in the control (Fig. 4, E and F). This result indicates that the SAIYAN system can detect the enhancement of the GTPase cycle of Sar1 by Sec13/Sec31. The physiological implications of these observations will be discussed later.

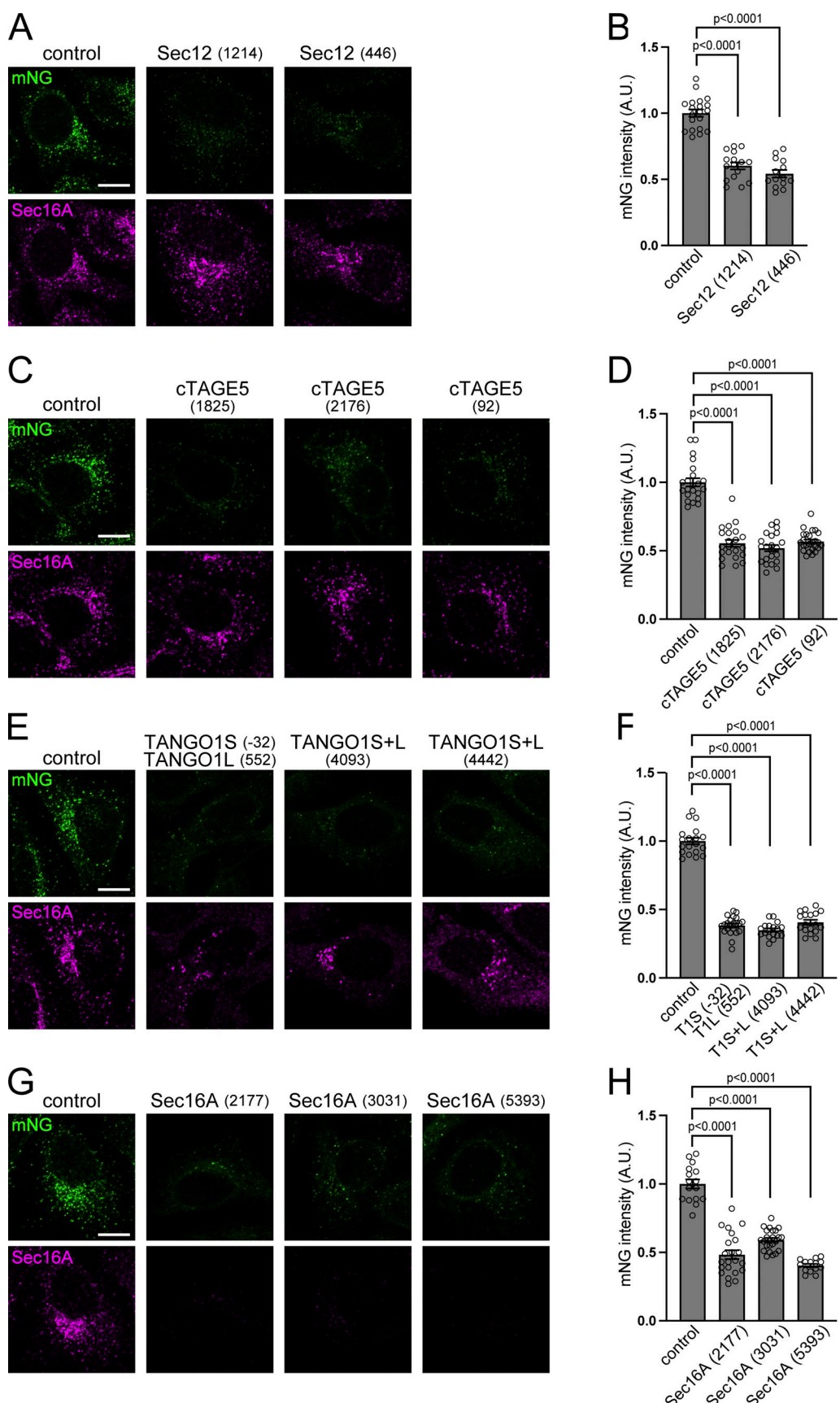

Figure 3. **ER exit site organization is required for the efficient activation of Sar1A. (A, C, E, and G)** Sar1A/SAIYAN (HeLa) cells transfected with the indicated siRNAs were fixed and stained with anti-Sec16-C antibodies. Scale bar = 10 μm. **(B, D, F, and H)** Quantification of mNG signals from A, C, E, and G, respectively (arbitrary units [A.U.]). Error bars represent the means ± SEM. Each data point represents the mNG intensity of the analyzed cells.

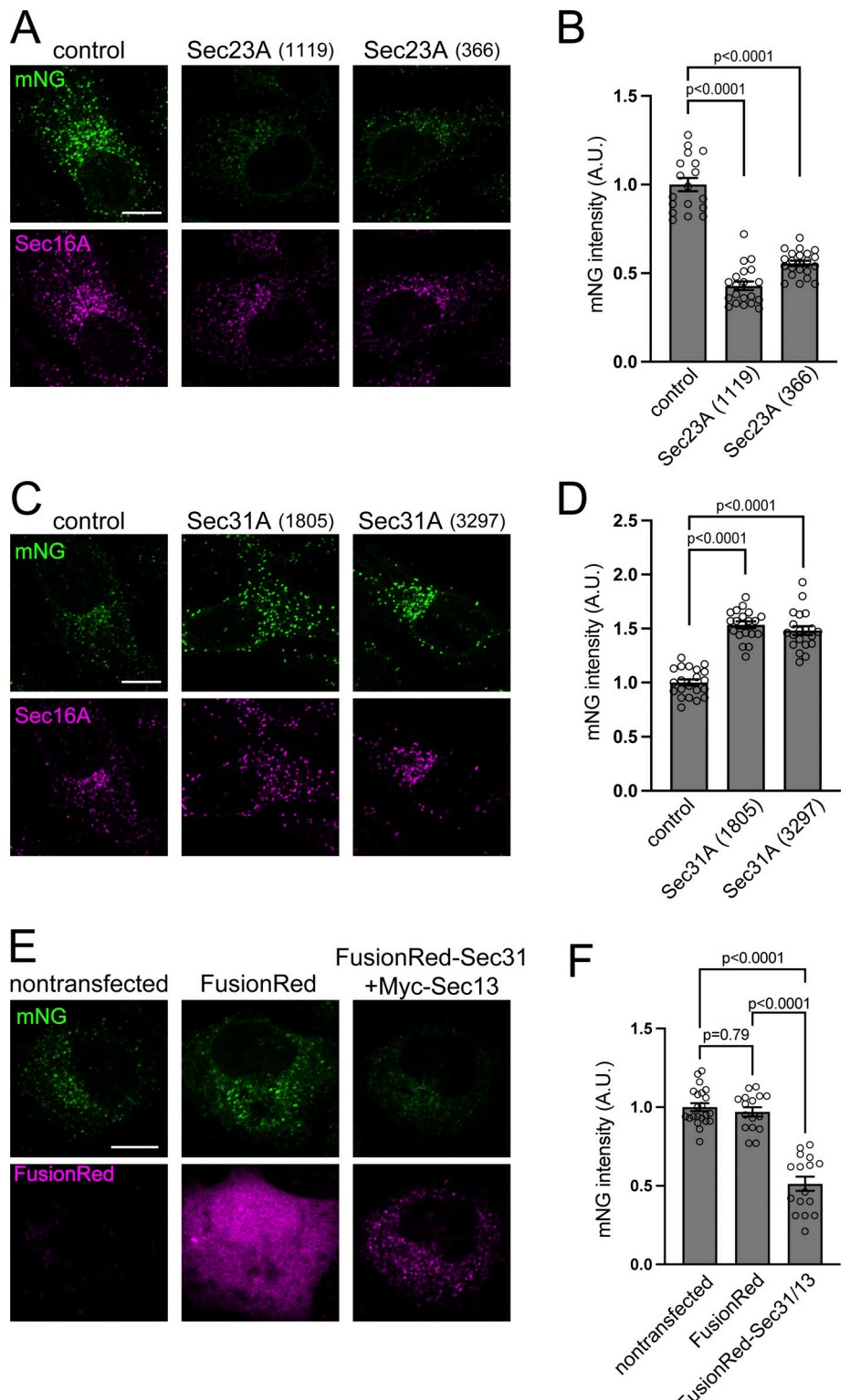

Figure 4. **Sec23A and Sec31A depletion exerts opposite effects on the activation of Sar1A in living cells. (A and C)** Sar1A/SAIYAN (HeLa) cells transfected with the indicated siRNAs were fixed and stained with anti-Sec16-C antibodies. Scale bar = 10 µm. **(B and D)** Quantification of mNG signals from A and C, respectively (arbitrary units [A.U.]). Error bars represent the means ± SEM. Each data point represents the mNG intensity of the analyzed cells. **(E)** Sar1A/SAIYAN (HeLa) cells transfected with the indicated plasmids were fixed and processed for microscopic analysis. Scale bar = 10 µm. **(F)** Quantification of mNG signals from E (A.U.). Error bars represent the means ± SEM. Each data point represents the mNG intensity of the analyzed cells.

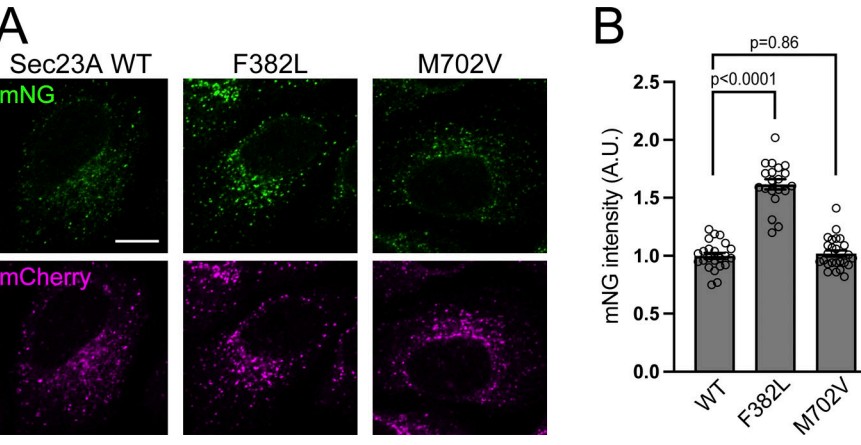

**A**

Sec23A WT | F382L | M702V
mNG
mCherry

**B**

Figure 5. **Each CLSD mutant of Sec23A exhibits different properties on Sar1 activation. (A)** Sar1A/SAIYAN (HeLa) cells stably expressing mCherry-tagged Sec23A constructs, as indicated, were fixed and processed for microscopic analysis. Scale bar = 10 μm. **(B)** Quantification of mNG signals from A (A.U.). Error bars represent the means ± SEM. Each data point represents the mNG intensity of the analyzed cells.

**Cranio-lenticulo-sutural dysplasia (CLSD) mutants of Sec23A showed activity against Sar1, consistent with in vitro studies**

CLSD (OMIM #607812) is a syndrome characterized by facial dysmorphism, late-closing fontanels, congenital cataracts, and skeletal defects caused by monoallelic or biallelic mutations in the Sec23A gene (Boyadjiev et al., 2006, 2011). Two mutations, F382L and M702V, have been extensively analyzed. While mutation in both alleles causes the F382L mutation to induce CLSD, M702V causes the condition even when only one allele is mutated. Discrepancies in the activities of both mutants toward Sar1A were investigated in vitro. Although neither mutation of Sec23A significantly affected Sar1A incubated as Sec23A/Sec24D in the absence of the Sec13/Sec31 complex, the presence of Sec13/Sec31 decreased the GAP activity toward Sar1A, particularly in the presence of the F382L mutation (Fromme et al., 2007; Kim et al., 2012). Therefore, using the SAIYAN system, we examined the impact of both mutations on living cells that contained endogenous Sec13/Sec31. Stable expression of various mCherry-tagged Sec23A mutants was achieved in SAIYAN cells; however, endogenous Sec23A expression was reduced, as shown in Fig. S3 G. The activation status of Sar1 was measured in these cells. mNG signals were significantly stronger in cells expressing the F382L mutation (Fig. 5 A). This observation was further supported by the quantitative results, which indicate a notable decrease in GAP activity toward Sar1A in cells expressing the F382L mutation (Fig. 5 B). Thus, SAIYAN cells allowed for the effective observation of various effects of Sar1 in live cells.

**Activated Sar1 prevailed in the ERGIC region of collagen-secreting cells**

Our results suggest that the SAIYAN system can detect the activation status of Sar1 in HeLa cells. We then performed detection of the activation status of Sar1 in collagen-secreting cells. Sar1A/SAIYAN cell lines were established from BJ-5ta cells, hTERT-immortalized fibroblasts of human foreskin origin (Bodnar et al., 1998; Jiang et al., 1999). Genomic sequencing confirmed the production of SAIYAN cells, which were designated as Sar1A/SAIYAN (BJ-5ta) cells. Cells treated with 2,2′-dipyridyl (DPD), a prolyl hydroxylase inhibitor known to impede collagen folding, exhibited augmented procollagen I accumulation within the ER, indicating collagen production and

secretion by the cell lines (Fig. S4) (Bonfanti et al., 1998; Mironov et al., 2003).

Subsequently, Sar1A/SAIYAN (BJ-5ta) cells were costained with Sec16. The cells were visualized using Airyscan microscopy, showing a 1.4-fold improvement in lateral resolution compared with traditional confocal imaging (Huff, 2015). Sec16 exhibited punctate localization within the cytoplasm, with a tendency to accumulate near the Golgi apparatus, indicating the characteristic localization of ER exit sites, as observed in HeLa cells (Fig. 6 A). Upon observing the mNG signal, the punctate localization showed a definite colocalization with Sec16 (Fig. 6 A). Notably, in addition to this pattern, the mNG signal exhibited weak extension in a reticular pattern, with less colocalization with Sec16 (Fig. 6 A). To further elucidate the nature of these membranous protrusions, we costained mNG signals with various antibodies targeting proteins localized at the ER–Golgi interface. This reticular pattern did not coincide with GM130 (Fig. 6 M) or PDI (Fig. 6 N), which are markers for the Golgi and ER, respectively. However, these reticular membranes containing mNG signals showed significant colocalization with the ERGIC53 antibodies (Fig. 6 B). Conversely, Rab1A, an ERGIC marker, exhibited limited colocalization with activated Sar1 (Fig. 6 O). Additionally, Sec23 (Fig. 6 C), Sec24B (Fig. 6 D), and p125A (Fig. 6 F), which are all inner COPII coat constituents, displayed substantial overlap with the reticular membranes of activated Sar1. The localization of Sec24D (Fig. 6 E) to the reticular signal in the ERGIC region was relatively weaker, and the degree of colocalization with mNG was not as pronounced as that with other inner coat components (Fig. 6 P).

The spatial relationships between these proteins were further elucidated using triple staining, as shown in Fig. 7. As demonstrated by dual staining, Sec16 overlapped with mNG in punctate structures, but there was minimal overlap with the reticular patterns of mNG (Fig. 7 A). As expected, ERGIC53 showed minimal colocalization with Sec16 (Fig. 7 A). In contrast, Sec23 not only overlapped with mNG but also colocalized with ERGIC53, demonstrating that Sec23 was localized not only at the ER exit sites but also across the ERGIC region (Fig. 7 B). Although ERGIC53 and Rab1A partially overlapped, they exhibited distinct regions, suggesting their combined involvement in shaping the ERGIC region (Fig. 7 C). mNG signals exhibited considerable overlap with ERGIC53 but minimal overlap with Rab1A, indicating

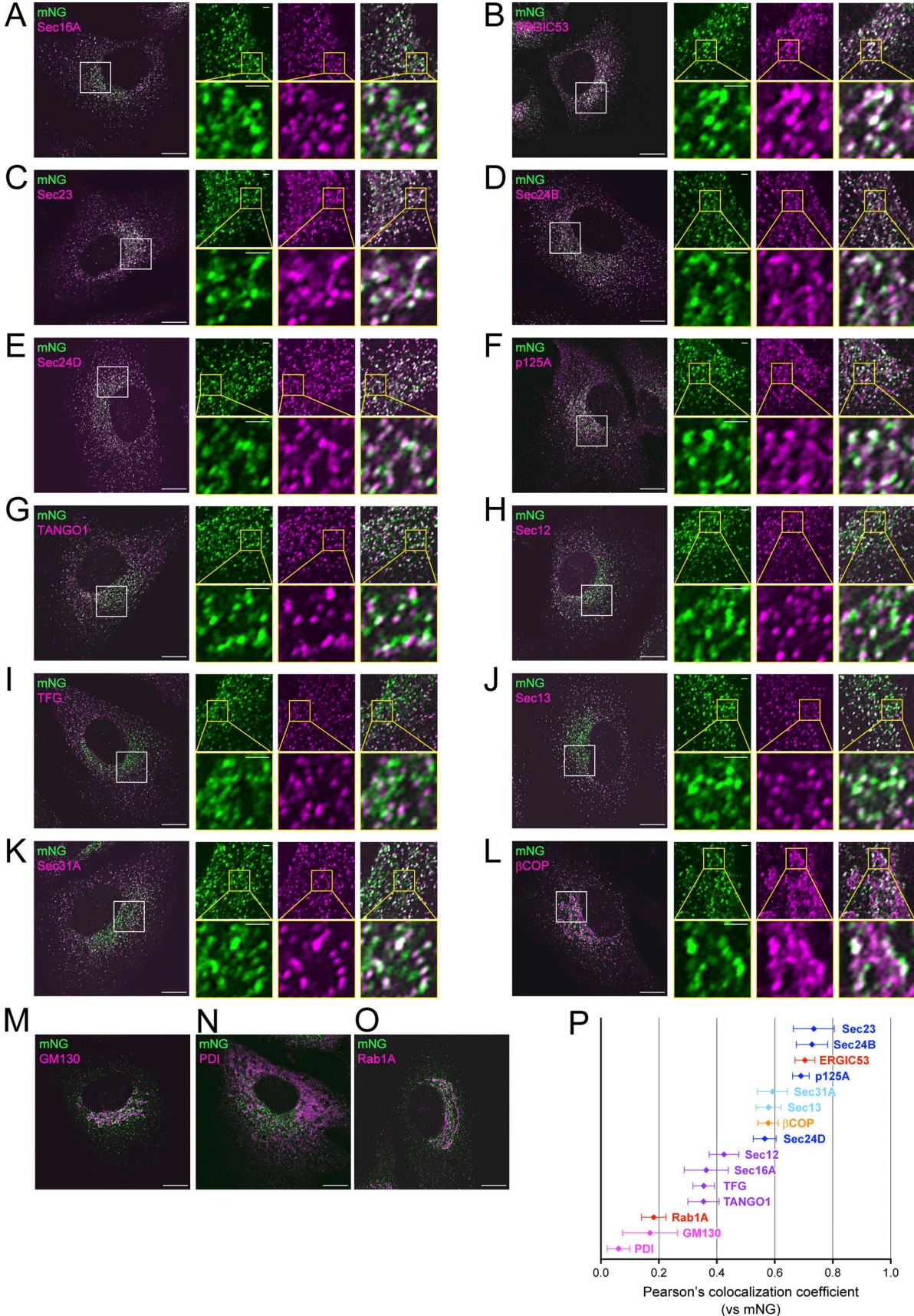

Figure 6. **Activated Sar1 prevails in the ERGIC region of Sar1A/SAIYAN (BJ-5ta) cells. (A–O)** Sar1A/SAIYAN (BJ-5ta) cells were fixed and stained with anti-Sec16-C (A), anti-ERGIC53 (B), anti-Sec23 (5H2) (C), anti-Sec24B (D), anti-Sec24D (E), anti-p125A (F), anti-TANGO1-CT (G), anti-Sec12 (H), anti-TFG (I), anti-

Sec13 (J), anti-Sec31A (mouse) (K), anti-β-COP (L), anti-GM130 (M), anti-PDI (N), and anti-Rab1A (O) antibodies. Images were captured using Airyscan2. Scale bars: 10 µm (main), 1 µm (magnification). **(A–L)** Right; top: Magnification of the indicated regions is on the left. Right; bottom: Magnification of the indicated regions on the upper. **(P)** Pearson's correlation coefficient was used to quantify the degree of colocalization. $n = 5$. Cyan; outer COPII coats, blue; inner COPII coats, purple; ER exit site resident proteins, red; ERGIC proteins, orange; COPI protein, magenta; ER and Golgi proteins. Error bars represent the mean 95% CI.

mNG localization to the ERGIC region where ERGIC53 predominated (Fig. 7 C).

In contrast, ER exit-site proteins, including TANGO1 (Fig. 6 G), Sec12 (Fig. 6 H), and TFG (Fig. 6 I), only exhibited colocalization with punctate structures similar to those of Sec16. Notable, Sec13 (Fig. 6 J) and Sec31A (Fig. 6 K), which form the outer COPII coats, exhibited less colocalization with mNG compared with the inner COPII coats, but more than the resident proteins of the ER exit sites. These results indicate that activated Sar1 predominated in the ERGIC regions in conjunction with the inner COPII components within collagen-secreting BJ-5ta cells. Conversely, proteins localized at the ER exit sites remain confined within the contiguous domain of the ER.

Finally, we examined Sec23A localization in parental BJ-5ta cells. As shown in Fig. 7 D, Sec23A extension was observed in the ERGIC region, confirming that the reticular signals observed in SAIYAN cells were not a result of introducing the SAIYAN system artificially. This observation further supports the extension of activated Sar1 to ERGIC regions in collagen-secreting BJ-5ta cells.

### Regions of Sar1 activation depended on the cargo secreted by the cell

The aforementioned observation aligns with the previous reports by Weigel et al. (2021) and Shomron et al. (2021), which described ER-Golgi protein transport mediated by an interconnected tubular network rather than spherical vesicles. Notably, their reports did not differentiate between resident proteins at ER exit sites and COPII coats (Shomron et al., 2021; Weigel et al., 2021). An intriguing finding in their studies was the involvement of COPI proteins in anterograde transport. We further explored the spatial relationship between COPI proteins and Sar1 activation in SAIYAN cells. Consistent with these previously reported findings, COPI proteins were found to be localized in close proximity to activated Sar1 (Fig. 6 L). Fig. 6 P illustrates the degree of colocalization of activated Sar1 with various factors present at the ER–Golgi interface in Sar1A/SAIYAN (BJ-5ta) cells. Inner COPII components, akin to ERGIC53, exhibited a high degree of colocalization with activated Sar1, followed by outer COPII components and COPI factors. The degree of colocalization with resident proteins present at ER exit sites, such as Sec12 responsible for activating Sar1, was significantly lower in comparison to the degree of colocalization with inner COPII components.

An intriguing comparison would involve examining the degree of colocalization between mNG signals and other components present at the ER–Golgi interface in Sar1A/SAIYAN (HeLa) cells (Fig. S5). In HeLa cells, the degree of colocalization of activated Sar1 was moderately high with ER exit-site resident proteins, similar to inner and outer COPII, while ERGIC and COPI exhibited comparatively lower colocalization. These results

suggest that the structure of the ER–Golgi interface varied significantly depending on the nature of the secreting cells.

Additionally, to validate this, we treated Sar1A/SAIYAN (BJ-5ta) cells with DPD and examined the change in the activation region of Sar1 upon the inhibition of collagen secretion. In cells treated with DPD, the previously observed weak reticular signal, which was a typical characteristic of BJ-5ta cells, disappeared, and the mNG signal localized to the punctate structures, similar to that observed in HeLa cells (Fig. 8 A). In fact, the measurement of colocalization with Sec16 showed a significant increase in DPD-treated cells compared with that in untreated cells (Fig. 8 B). These findings imply that the ER–Golgi interface adopts different structures depending on the cargo being secreted, thereby maintaining a shape customized for the specific cargo. Furthermore, it became evident that activated Sar1 in collagen-secreting cells was localized along with the inner COPII components from the ER exit site to the ERGIC region.

## Discussion

In this study, we successfully visualized the activation of Sar1 in living cells for the first time using split fluorescent protein technology, designated as the SAIYAN system. Previously, the measurement of Sar1 activation relied solely on in vitro biochemical assays. However, SAIYAN technology accurately identified the activity of Sar1 modulators in living mammalian cells. Specifically, we directly measured the requirement of Sec12 as a GEF and cTAGE5 as a recruiter for Sec12 to activate Sar1 in living cells. Previously, we had considered the importance of Sar1 activation at the ER exit site based on two key points: (1) Sec12 accumulates at the ER exit site in a cTAGE5-dependent manner (Saito et al., 2014), and (2) the function of cTAGE5, which cannot bind to Sec12, is maintained by the overexpression of Sar1 WT (Tanabe et al., 2016). For the first time, we could detect the actual occurrence of Sar1 activation at the ER exit site using the SAIYAN system. Furthermore, we elucidated the roles of essential factors, such as TANGO1 and Sec16, which are components of ER exit sites, in facilitating appropriate Sar1 activation (Tanabe et al., 2016).

Sec23 has been considered a dual-function protein because it exhibits GAP activity toward Sar1 in vitro (Yoshihisa et al., 1993) and, together with Sec24, it binds activated Sar1 and cargo receptors bound to Sec24 to form a prebudding complex (Bi et al., 2002; Kuehn et al., 1998; Sato, 2004). Additionally, Sec13/Sec31 enhances the GAP activity of Sec23 toward Sar1 by about 10-fold (Bi et al., 2007). To date, the implications of these findings in mammalian cells have not been clearly elucidated. In this study, we observed that in Sec23 knockdown cells, activated Sar1 was downregulated. Conversely, the knockdown of Sec13/Sec31 upregulated the activated Sar1. These results suggest that Sec23/Sec24 alone do not function as GAPs within cells but rather play

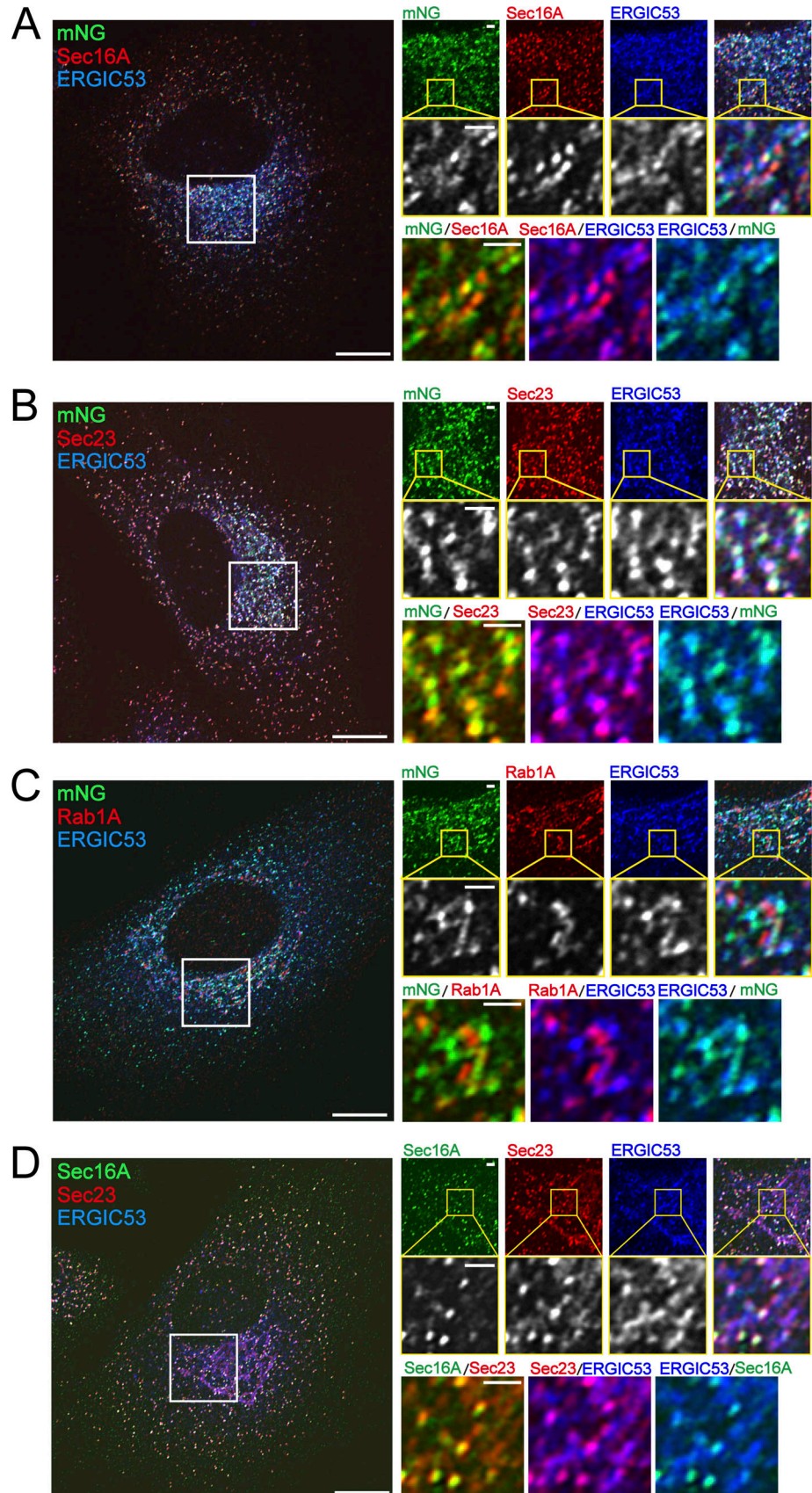

Figure 7.    **Triple staining of Sar1A/SAIYAN (BJ-5ta) and parental BJ-5ta reveals the organization of the ER-Golgi interface of collagen-secreting cells.**
**(A–C)** Sar1A/SAIYAN (BJ-5ta) cells were fixed and stained with anti-Sec16-C and anti-ERGIC53 (A), anti-Sec23 (5H2), and anti-ERGIC53 (B), and anti-Rab1A and

anti-ERGIC53 (C) antibodies. Images were captured using the Airyscan2. Scale bars: 10 µm (main), 1 µm (magnification). **(D)** BJ-5ta cells were fixed and stained with anti-Sec16-C, anti-Sec23 (5H2), and anti-ERGIC53 antibodies. Images were captured using the Airyscan2. Scale bars: 10 µm (main), 1 µm (magnification). **(A–D)** (right; top) Magnification of the indicated regions is on the left. (right; bottom) Double staining of the magnified region on the top.

a role in the formation of the prebudding complex. Only in the presence of Sec13/Sec31 does Sec23/Sec24 undergo a conformational change, allowing them to exhibit sufficient GAP activity for Sar1 hydrolysis (Bi et al., 2007). Moreover, the decrease in the mNG signal detected by SAIYAN upon Sec13/Sec31 overexpression indicates that the Sar1-FLAG-mNG$_{11}$ and HA-mNG$_{1-10}$ complex formation is not irreversible and can be disrupted by Sec13/Sec31. We also obtained conclusions consistent with previous in vitro results regarding the activity of well-analyzed CLSD-associated mutants of Sec23A, F382L, and M702V on Sar1A. The M702V mutant has been reported to have more substantial effects on Sar1B than on Sar1A (Kim et al., 2012). Further development and analysis with SAIYAN cells for Sar1B are anticipated to enhance our understanding of CLSD within cells. We devised a novel system employing SAIYAN technology, allowing real-time monitoring of the activation of small GTPases, such as Sar1, whose effector also functions as a GAP in live cells—a capability unattainable with previous FRET probes.

Unexpectedly, activated Sar1 seemed not confined solely to the ER exit site but rather extended beyond the ER exit site into the ERGIC region in collagen-secreting cells, a phenomenon not observed in HeLa cells secreting minimal collagen. Notably, upon the addition of DPD, a collagen folding inhibitor, activated Sar1 was specifically localized to the ER exit site, even in BJ-5ta cells. These results imply that the specific localization of activated Sar1 to the ERGIC region might be crucial for collagen secretion. Moreover, the site of Sar1 activation and its function may differ during the process of collagen secretion. Based on the

insights gained from the activation of Sar1 and collagen transport from the ER, reports have indicated that Sar1 H79G addition to semi-intact cells results in tubular structures that emanate from the ER (Aridor et al., 2001). Similar tubular formation has been observed when Sar1 H79G is introduced to artificial liposomes or incubated with non-hydrolyzable GTP analogs, such as GTPγS and GMP-PNP (Antonny et al., 2001; Bacia et al., 2011; Bielli et al., 2005; Long et al., 2010). These findings are consistent with studies that have reported that upon Sar1 activation, the amphipathic N-terminal region of Sar1 becomes embedded in the membrane, potentially inducing membrane curvature (Bi et al., 2002; Paul et al., 2023). Additionally, using cryoelectron microscopy, Zanetti et al. (2013) demonstrated that non-hydrolyzable Sar1 and COPII components incubated with giant unilamellar vesicles induce the formation of tubes covered by Sec23/Sec24 and Sec13/Sec31. Remarkably, their model predicted that the tubular structures covered with Sec23/Sec24 would recruit fewer Sec13/Sec31 than the spherical vesicles. Additionally, an enlarged COPII cage suggested a decrease in Sar1-GTP hydrolytic activity compared with conventional COPII coats. We found that upon the secretion of collagen, activated Sar1 extended from the ER exit site to the ERGIC region, where Sec23/Sec24 and p125 colocalized. This observation is consistent with the tubular structures identified by Zanetti et al. (2013) upon the addition of GTP-locked Sar1 in vitro. Additionally, the colocalization of Sec13/Sec31 with activated Sar1 was lower than that of inner coat factors but higher than that of ER exit site–resident proteins, thus supporting their proposed model. However, the low degree of colocalization between the activated

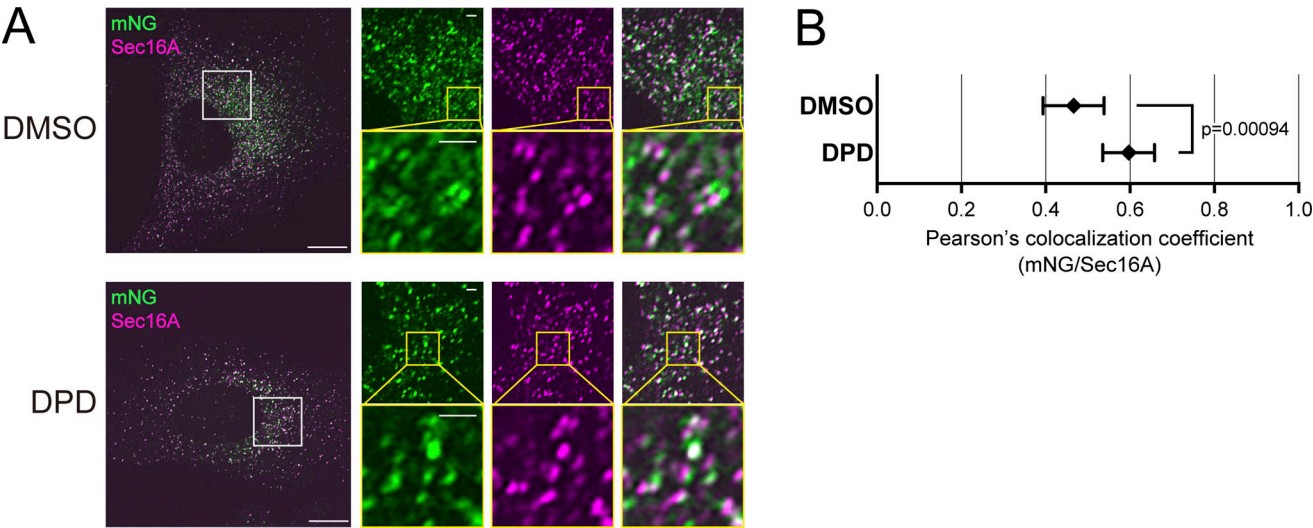

**Figure 8.  Reticular pattern of activated Sar1 signals diminished with DPD treatment in Sar1A/SAIYAN (BJ-5ta) cells. (A)** Sar1A/SAIYAN (BJ-5ta) cells were treated with DMSO or 0.5 mM DPD and incubated for 16 h. Cells were fixed and stained with anti-Sec16-C antibodies. Images were captured using the Airyscan2. Scale bars: 10 µm (main), 1 µm (magnification). **(B)** Pearson's correlation coefficient was quantified to assess the degree of colocalization. *n* = 5. Error bars represent the mean 95% CI.

Sar1 indicated by the SAIYAN mNG signal and Sec13/Sec31 could also be due to the efficient inactivation of Sar1 by Sec13/Sec31. This possibility cannot be ruled out and will require further investigation, including understanding the functions of activated Sar1 in the ERGIC region. Mutations in Sec24D (Fig. 6 E) cause osteogenesis imperfecta (OI), suggesting that Sec24D is involved in collagen secretion (Garbes et al., 2015). Therefore, the observation that Sec24D is less colocalized with activated Sar1 in ERGIC protrusions in collagen-secreting cells was unexpected. However, evidence suggests that collagen accumulation is not significant in Sec24D-deficient patient skin fibroblasts and that all Sec24 isoforms are involved in collagen transport, indicating the need for further analysis (Garbes et al., 2015; Lu et al., 2022).

Furthermore, Santos et al. (2015) proposed the possibility of ERGIC membrane recruitment to TANGO1 located at ER exit sites, suggesting the formation of large carriers necessary for collagen secretion. Subsequently, they suggested a model in which elongated tunnel-like structures facilitate the secretion of collagen from the ER to the Golgi apparatus (Bunel et al., 2024; Raote and Malhotra, 2019). Recent observations using live cells and electron microscopy have revealed reticular structures at the ER–Golgi interface, indicating that protein transport may occur through these reticular/tubular structures instead of vesicular transport alone. Nevertheless, these studies primarily focused on the localization of inner and outer COPII proteins as indicative of proteins at the ER–Golgi interface, which may not have clearly distinguished between the resident proteins at the ER exit sites, COPII proteins, and proteins localized at the ERGIC structures. Moreover, most of these analyses were confined to cells, such as HeLa, COS7, and CHO, which are known to secrete limited amounts of collagen (Shomron et al., 2021; Weigel et al., 2021). Our observations suggest that, at least in collagen-secreting cells, resident proteins at ER exit sites and COPII proteins may behave differently depending on the cell type. Thus, our current analysis provides some modifications to their observations. A schematic representation of the ER–Golgi interface is shown in Fig. 9. Factors, such as TANGO1, Sec16, Sec12, and TFG, were localized specifically at the punctate structures commonly referred to as ER exit sites in collagen-secreting cells, with no evident reticular structures directed toward ERGIC (Fig. 9 B). Conversely, inner coat proteins composed of Sec23/Sec24 and p125A protruded toward the ERGIC direction, along with activated Sar1, in addition to their localization at conventional ER exit sites (Fig. 9 B). Activated Sar1 was restricted to the ERGIC53-positive region of the ERGIC and was absent from the Rab1 side of these cells (Fig. 9 B). This observation correlates with a report indicating that Surf4 is present in tubular ERGIC, which is Rab1-positive and ERGIC53-negative (Yan et al., 2022). Furthermore, recent discoveries highlighting the interaction of p125 with VPS13B and its role in collagen secretion via tubular ERGIC formation underscore its significance (Du et al., 2024, Preprint). Moreover, our discovery that activated Sar1 in collagen-secreting cells demonstrates stronger colocalization with β-COP than ER exit site proteins is notable (Fig. 6 P). This observation aligns with prior evidence implicating COPI in collagen secretion (Raote et al., 2018). In contrast, in HeLa

cells, both resident and COPII proteins were found to colocalize well with activated Sar1 at the ER exit sites, while ERGIC and COPI exhibited less colocalization with activated Sar1 (Fig. 9 A).

Our findings are consistent with both the vesicle-mediated and tunnel-mediated transport models. It remains a challenge to determine whether all secretory proteins are exclusively transported via reticular structures or if both vesicular transport and reticular structures coexist (Phuyal and Farhan, 2021). In this context, whether the differences in the structure of the ER–Golgi interface can be attributed to the mode of secretion is an intriguing issue. The mechanism by which activated Sar1 is stabilized and facilitates protrusions into the ERGIC in collagen-secreting cells compared with minimal collagen-secreting cells remains a subject for future investigation. Additionally, variations in the structure of the ER–Golgi interface across different cell types suggest the flexibility of the ER–Golgi interface to accommodate cargo types specific to each cell type. Indeed, in our experiments, treatment of collagen-secreting cells with DPD resulted in their ER–Golgi interfaces resembling those in HeLa cells. Recent studies indicating normal secretion in cells lacking Sar1 and the role of mechanical strain in regulating secretion via Sar1 through Rac1 activity emphasize the importance of elucidating the role of Sar1 in secretion (Kasberg et al., 2023; Phuyal et al., 2022). Thus, understanding the role of Sar1 in vesicular secretion is critical for future research.

One limitation of the current approach is the inherent tendency of split fluorescent proteins to self-associate to some extent (Romei and Boxer, 2019). While split fluorescent proteins have been successfully utilized as factors forming membrane contact sites, improper levels of expressed factors can lead to the formation of artificial contact sites (Tashiro et al., 2020). During the process of initial optimization of the system, when we lengthened the glycine linker attached to Sar1, we observed mNG signals by SAIYAN, even in ΔN Sar1 mutants that were not recruited to the membranes (data not shown).

In addition, the SAIYAN system measured the membrane-bound state of Sar1 rather than its nucleotide form. Nonetheless, as GDP-bound Sar1 binds to membranes more weakly than GTP-bound Sar1 (Paul et al., 2023), our validation of the SAIYAN system showed that the T39N Sar1 mutant exhibited approximately half the mNG signal compared with that of the WT Sar1 (Fig. 1 C). This result is consistent with the mNG signal levels observed when Sec12 is knocked down (Fig. 3 B) or when Sec13/Sec31 is overexpressed (Fig. 4 F). Thus, while the SAIYAN system measures the amount of membrane-bound Sar1, it can still detect shifts in Sar1's nucleotide form based on whether the mNG signal increases or decreases following a particular treatment. Therefore, when applying SAIYAN to other small GTPases, it is crucial to measure the SAIYAN signal for both the GDP-bound and GTP-bound forms using mutants in advance. This will validate the range of mNG signals that can accurately reflect changes in the nucleotide form. Despite these requirements, SAIYAN offers an advantage over existing relocation biosensors, where any fluorescence in the cytoplasm that is not bound to the activated GTPase is considered background. In contrast, SAIYAN is expected to reduce background fluorescence

Figure 9.  **Schematic representation of ER-Golgi interfaces observed with SAIYAN system. (A)** In HeLa cells, ER exit site resident proteins exhibit strong colocalization with inner and outer COPII coat proteins at the ER exit sites. In contrast, ERGIC is located away from the ER exit sites. **(B)** In collagen-secreting BJ-5ta cells, although ER exit site resident proteins are confined to the ER exit sites, inner COPII coats extend to the ERGIC regions labeled with ERGIC-53 but not Rab1. Outer coat proteins partially colocalize with ER exit sites.

because it only emits fluorescence when the small GTPase is membrane-bound.

In future iterations of SAIYAN, the use of more transiently interacting fluorescent probes will be beneficial. Nevertheless, in this study, we successfully detected Sar1 activation in cells, for the first time, using split fluorescent proteins and gained intriguing insights into the differential activation states of Sar1 across different cell types. By further refining the SAIYAN technology, we anticipate gaining broader insights into the intracellular activation of small GTPases.

## Materials and methods

### Antibodies
Polyclonal antibodies against Sec13 (RRID: N/A) were raised in rabbits by immunization with His$_6$-Sec13 and affinity-purified by columns conjugated with GST-Sec13. Polyclonal antibodies against TFG (RRID: N/A) were raised in rabbits by immunization with TFG (308–396 aa) and affinity-purified by columns conjugated with corresponding region of TFG. Polyclonal antibodies against TANGO1-CC1 (RRID: N/A, rabbit) or cTAGE5-CC1 (RRID: N/A, rabbit) were raised in rabbits by immunization with recombinant GST-tagged TANGO1 (1,231–1,340 aa) or cTAGE5 (118–227 aa) and affinity-purified by columns conjugated with ColdTF-tagged TANGO1 (1,231–1,340 aa) or MBP-tagged cTAGE5 (118–227 aa; Maeda et al., 2016; Saito et al., 2011; Tanabe et al., 2016). Polyclonal antibody against Sec31A (RRID: N/A, rabbit) was raised in rabbits by immunization with recombinant Sec31A (aa 522–719) and affinity-purified by columns conjugated with GST-tagged Sec31A (aa 522–719; Maeda et al., 2020). Polyclonal antibodies against Sec16-N (374–387 aa; RRID: N/A, rabbit),

Sec16-C (2,319–2,332 aa; RRID: N/A, rabbit), and TANGO1-CT (1,884–1,898 aa; RRID: N/A, rabbit) were raised in rabbits by immunization with keyhole limpet hemocyanin-conjugated peptides and affinity-purified by columns conjugated with the peptides (Thermo Fisher Scientific; Maeda et al., 2016; Saito et al., 2009; Saito et al., 2011; Tanabe et al., 2016). For the production of anti-Sec23 (RRID: N/A, rat) and anti-Sec12 (RRID: N/A, rat) antibodies, a 6-wk-old female Wistar rat (CLEA Japan) was immunized with FLAG-tagged Sec23A or GST-tagged Sec12 (93–239 aa) in TiterMax Gold (TiterMax USA). Splenocytes were fused with PAI mouse myeloma cells using polyethylene glycol (Roche). Hybridoma supernatants were screened by indirect ELISA with His-tagged Sec23A or ColdTF-tagged Sec12 (93–239 aa) as the antigens. Positive hybridoma lines were subcloned, grown in serum-free medium (Nihon Pharmaceutical) supplemented with hypoxanthine–thymidine (Thermo Fisher Scientific), and purified with protein G-Sepharose (GE Healthcare; Maeda et al., 2017; Saito et al., 2014). The Sec23 antibody (5H2) recognizes both Sec23A and Sec23B, whereas the Sec23A antibody (11D8) specifically recognizes Sec23A. The Sec24D antibody (RRID: N/A) was gifted by Schekman Lab. Other antibodies used in this study were as follows: anti-GAPDH (Cat# sc-32233, RRID:AB_627679; Santa Cruz Biotechnology), anti-FLAG (Cat# F1804, RRID:AB_262044; Sigma-Aldrich) or anti-FLAG (Cat# 200473, RRID:AB_10596510; Agilent), anti-HA (Cat# 11867423001, RRID:AB_390918; Roche), anti-Sar1A (Cat# 22291-1-AP, RRID:AB_2879062; Proteintech), anti-ERK1 (Cat# sc-271269, RRID:AB_10611091; Santa Cruz Biotechnology), anti-calnexin (Cat# 610523, RRID:AB_397883; BD Biosciences), anti-Sec31A (Cat# 612350, RRID:AB_399716; BD Biosciences), anti-Sec24B (Cat# 12042, RRID:AB_2797807; Cell Signaling Technology),

anti-p125A (Sec23IP, Cat# 20892-1-AP, RRID:AB_10896458; Proteintech), anti-collagen I (Cat# SP1.D8, RRID:AB_528438; DSHB), anti-β-COP (Cat# G6160, RRID:AB_477023; Sigma-Aldrich), anti-ERGIC53 (Cat# sc-398893, RRID:AB_2905549; Santa Cruz Biotechnology), anti-Rab1a (Cat# 13075, RRID:AB_2665537; Cell Signaling Technology), anti-GM130 (Cat# 610823, RRID:AB_398142; BD Biosciences), and anti-PDI (Cat# ab2792, RRID:AB_303304; Abcam).

## Constructs

For the transient expression of human Sar1A WT or mutant constructs, the human Sar1A cDNA fused to the C terminal with the FLAG peptide, GS linker (GGGS), and the 11th β-strand (214–229 aa) of mNeonGreen2 was cloned into pCMV5 vectors that were gifted by D. Russell (RRID:N/A, University of Texas Southwestern Medical Center, Dallas, TX, USA) (Feng et al., 2017). The ΔN mutant lacks the 25aa of the N terminal, which corresponds to the 23aa deletion of yeast constructs (Lee et al., 2005).

## Cell culture and transfection

HeLa cells (Cat# 300194/p772_HeLa, RRID:CVCL_0030; CLS) were cultured in DMEM supplemented with 10% FBS. BJ-5ta cells (ATCC:CRL-4001) were cultured in high-glucose DMEM supplemented with Medium 199, 10% FBS, and ascorbic acid phosphate. For transfecting siRNA, Lipofectamine RNAi max (Thermo Fisher Scientific) was used. For single siRNA transfection, the reverse transfection protocol was used. In the case of double siRNA transfection, after 24 h of reverse transfection, forward transfection was conducted according to the manufacturer's protocol. For plasmid transfection, Fugene 6 (Promega) or Fugene 4K (Promega) was used. Doxycycline-inducible stable cell lines expressing HA-mNG$_{1-10}$ were made with the lentivirus system described previously (Shin et al., 2006). HA-mNG$_{1-10}$ constructs consisted of 1–92 aa of TANGO1S (transmembrane region of TANGO1S), HA peptide, and 1–213 aa (1–10th β-strands) of mNeonGreen2. Proteins were induced by incubation with 1 µg/ml doxycycline for 24 h. Stable cell lines expressing mCherry-Sec23A WT, F382L, and M702V were prepared with lentivirus by replacing Cas9 in LentiCRISPRv2 with Sec23A cDNAs (Sanjana et al., 2014). 2,2′-dipyridyl was purchased from TCI.

## CRISPR-mediated knock-in

sgRNA was prepared essentially as previously described (Feng et al., 2017; Leonetti et al., 2016; Lin et al., 2014). The DNA template for sgRNA was generated by PrimeSTAR GXL DNA Polymerase (TAKARA) by overlapping PCR using a set of three primers: BS7R: 5′-AAAAAAAGCACCGACTCGGTGC-3′ and ML611R: 5′-AAAAAAAGCACCGACTCGGTGCCACTTTTTCAAGTTGATAACGGACTAGCCTTATTTAAACTTGCTATGCTGTTTCCAGCATAGCTCTTAAAC-3′ and Sar1a_sgRNA_F: 5′-CCTCTAATACGACTCACTATAGGCAATATACTGGGAGAGCCAGGTTTAAGAGCTATGCTGGAA-3′. In vitro transcription and sgRNA purification were performed by CUGA7 gRNA Synthesis Kit (NIPPON Gene). To construct the donor plasmid for homology-directed repair, the Sar1 genome template was first amplified from HeLa genomic DNA with the following primers (5′-CCGCTCTAGAACTAGTAC

CCAAATGAGCTCTGGC-3′, and 5′-CGGTATCGATAAGCTTGCATCAGTATTAAATACACATG-3′) and cloned into pBSIISK(–) using the In-Fusion HD cloning Kit (TAKARA). The BamHI sequence was then inserted into the middle of the genomic Sar1 using the following primers (5′-TCCTTGGACGGTGAAAATAAAAGAGTTTTACTTC-3′, and 5′-TCCGTCAATATACTGGGAGAGCCAGCGG-3′). The FLAG peptide, GS linker (GGGS), and 214–229 aa of mNeonGreen2 corresponding to the 11th strand of mNeonGreen2 were then inserted into the BamHI locus to express Sar1A as a fusion with these sequences (Feng et al., 2017). The construct was then used as a donor template for ssDNA production by Guide-it Long ssDNA production system v2 (TAKARA). The primers 5′-GCATCAGTATTAAATACACATG-3′ and 5′-ACCCAAATGAGCTCTGGCCTCCATATC-3′ with or without 5′ phosphorylation were used for amplifying dsDNA for Strandase reaction to make ssDNA. For CRISPR/Cas9-mediated cell line generation of SAIYAN cells, 1 × 10$^5$ HeLa cells or BJ-5ta cells expressing inducible HA-mNG$_{1-10}$ were transfected with 7.5 pmol of Cas9 (TAKARA), 7.5 pmol of sgRNA, and 0.64 pmol of ssDNA by Nucleofector II (Lonza) or Neon Transfection System (Thermo Fisher Scientific). Cas9 and sgRNA were incubated at room temperature for 20 min prior to transfection. Electroporated cells were cultured in 24-well plates for 5–7 days and transferred to six-well plates prior to selection by fluorescence-activated single-cell sorting using FACS Melody (BD Biosciences).

## Cell viability assay

The cell viability of the cultured cells was quantified using the Cell Counting Kit 8 (DOJINDO Laboratories).

## RUSH assay

Sar1A/SAIYAN (HeLa) cells on a 3.5-cm glass bottom dish (Matsunami) treated with or without doxycycline were transfected with Str-KDEL_ManII-SBP-mCherry (Addgene). After 24 h of transfection, the medium was replaced with phenol red-free DMEM and treated with 40 µM biotin, and live imaging was performed by confocal laser scanning microscopy (Plan Apochromat 63×/1.40 NA oil immersion objective lens; LSM 900; Carl Zeiss) in a CO$_2$ incubator (Stage top incubator; TOKAI HIT) at 37°C. The acquired images were processed with Zen Blue software (RRID:SCR_013672; Carl Zeiss).

## Cell fractionation assay

Sar1A/SAIYAN (HeLa) cells treated with or without doxycycline were suspended with 300 µl of extraction buffer consisting of 320 mM sucrose, 20 mM Tris-HCl (pH 7.4), and 1 mM EDTA and passed through 23-gauge syringes 10 times. This was repeated with 30-gauge needles before the solution was centrifuged at 900 × g for 10 min at 4°C. The lysates were then centrifuged at 100,000 × g for 30 min at 4°C. The pellets were washed with PBS twice and resuspended with extraction buffer. The supernatants and pellets were processed for sample preparation for SDS–PAGE, followed by western blotting.

## Western blotting

The experiments were performed as described previously (Saito et al., 2014). Cells extracted with extraction buffer consisting of

20 mM Tris-HCl (pH 7.4), 100 mM NaCl, 1 mM EDTA, 1% Triton X-100, and protease inhibitors were centrifuged at 20,000 × $g$ for 15 min at 4°C. The blots were analyzed using ImageQuant 800 (RRID:SCR_014246; GE Healthcare). All western blotting figures were representative of at least three individual experiments.

### siRNA oligos
The siRNA oligos used in this study are shown in Table S1. The number in the parentheses represents the starting base pair of the target sequence. For control siRNA, Stealth RNAi siRNA Negative Control Med GC Duplex #2 (Thermo Fisher Scientific) was used.

### Immunofluorescence microscopy
Immunofluorescence microscopy analysis was performed as described previously (Saito et al., 2014). Cells grown on coverslips were washed with PBS, fixed with either methanol (6 min at –20°C) or 4% PFA/PBS (10 min at room temperature), and then washed with PBS and blocked in blocking solution (5% BSA in PBS with 0.1% Triton X-100 for 15 min). After blocking, cells were stained with primary antibody (1 h at room temperature), followed by incubation with Alexa Fluor-conjugated secondary antibodies (Alexa Fluor 488, Alexa Fluor 568, and/or Alexa Fluor 647 for 1 h at room temperature). The coverslips were mounted with PermaFluor Aqueous Mounting Medium (Epredia). Images were acquired with confocal laser scanning microscopy (Plan Apochromat 63×/1.40 NA oil immersion objective lens; LSM 900; Carl Zeiss). The acquired images were processed with Zen Blue software (RRID:SCR_013672, Carl Zeiss). All imaging was performed at room temperature. All immunofluorescence figures were representative of at least three individual experiments. mNeonGreen intensity scanning was performed using Fiji-ImageJ (Schindelin et al., 2012). Pearson's colocalization coefficient was analyzed using Zen Blue software (RRID:SCR_013672; Carl Zeiss).

### Statistical analysis
Data were analyzed using an unpaired two-sample Student's $t$ test for two-group comparisons or a one-way analysis of variance followed by Dunnett's test for multiple comparisons using GraphPad Prism software (RRID:SCR_002798; GraphPad). Data distribution was assumed to be normal, but this was not formally tested. All graphs were created using GraphPad Prism software (RRID:SCR_002798; GraphPad).

### Online supplemental material
The additional images regarding the validation of SAIYAN technology are presented in Fig. S1 (related to Fig. 1). Fig. S2 demonstrates that Sar1A/SAIYAN (HeLa) cells exhibited normal proliferation and that Golgi structure and secretion were also normal (related to Fig. 2). Fig. S3 shows western blotting results for Sar1A/SAIYAN (HeLa) cells subjected to various knockdowns and transfections (related to Figs. 3, 4, and 5). Fig. S4 confirms the accumulation of collagen in BJ-5ta cells treated with DPD (related to Fig. 8). Fig. S5 depicts the degree of colocalization of mNG and various organelle markers in Sar1A/SAIYAN (HeLa) cells (related to Fig. 6 P). Table S1 shows the siRNA sequences used in this study.

### Data availability
The data are available from the corresponding author upon reasonable request.

## Acknowledgments
This work was supported by the Japan Society for the Promotion of Science Grants-in-Aid for Scientific Research (20K15740, 22H02760, 23K24023 to M. Maeda, 24K18067 to M. Arakawa, 23H05254 to Y. Komatsu, and 19K22612, 20H03203, 20H04897, 21K19470, 23H02430, 23K27123 to K. Saito) from the Ministry of Education, Culture, Sports, Science and Technology of Japan. The work was further supported by the Akita University Support for Fostering Research Project to M. Maeda, Y. Komatsu, and K. Saito, by the Naito Foundation to K. Saito and M. Maeda, by the Takeda Science Foundation to K. Saito, by the Toray Science Foundation (19-6005) to K. Saito, by the Sumitomo Foundation to K. Saito, by the Suzuken Memorial Foundation to K. Saito, by the Yasuda Medical Foundation to K. Saito, by the Foundation for Promotion of Cancer Research to K. Saito, by the Asahi Glass Foundation to K. Saito, by the Princess Takamatsu Cancer Research Foundation to K. Saito, by the Japan Foundation for Applied Enzymology to M. Maeda, by the Kato Memorial Bioscience Foundation to M. Maeda, by the Astellas Foundation for Research on Metabolic Disorders to M. Maeda, by the Inamori Foundation to M. Maeda, by the Kao Foundation for Arts and Sciences to M. Maeda, and by the Koyanagi Foundation to M. Maeda. Open Access funding provided by Akita University.

Author contributions: M. Maeda: Conceptualization, Data curation, Formal analysis, Funding acquisition, Investigation, Methodology, Resources, Validation, Visualization, Writing—original draft, Writing—review & editing, M. Arakawa: Funding acquisition, Investigation, Resources, Writing—review & editing, Y. Komatsu: Funding acquisition, Investigation, K. Saito: Conceptualization, Data curation, Formal analysis, Funding acquisition, Investigation, Methodology, Project administration, Resources, Supervision, Validation, Visualization, Writing—original draft, Writing—review & editing.

Disclosures: The authors declare no competing interests exist.

Submitted: 28 March 2024

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

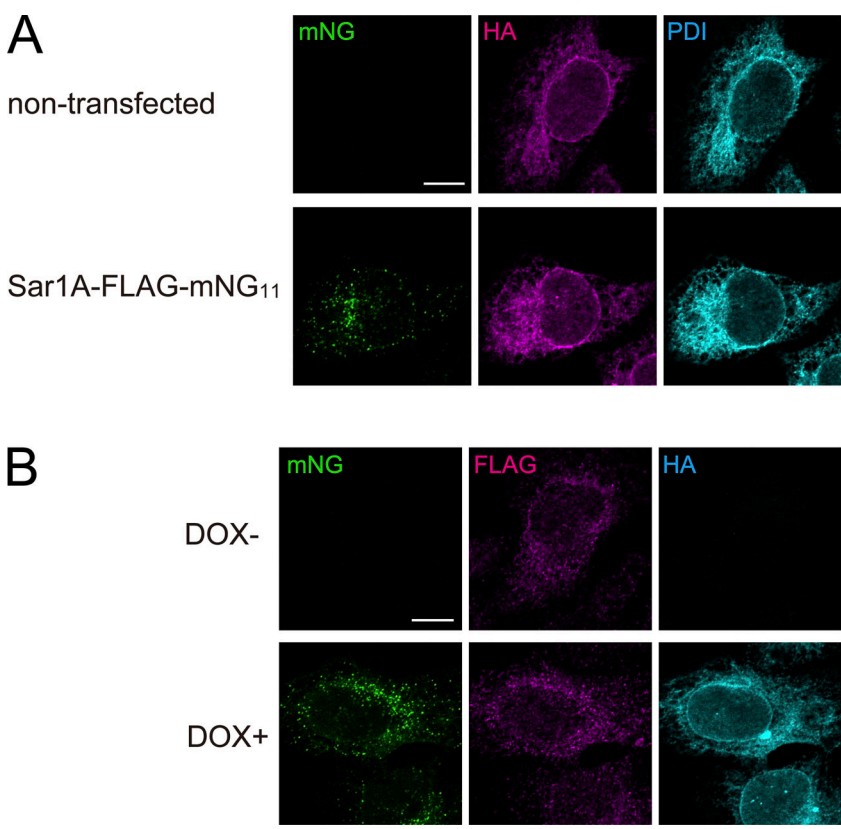

Figure S1. **Validation of SAIYAN technology. (A)** Doxycycline-inducible HeLa cells expressing the membrane-spanning region of TANGO1S and HA-tag fused to 10 of the 11 strands of mNG (HA-mNG$_{1-10}$ cells) were either non-transfected or transfected with Sar1A constructs with a FLAG tag and a glycine linker fused to the 11th strand of mNG (Sar1A-FLAG-mNG$_{11}$). The cells were fixed and stained with anti-HA and anti-PDI antibodies. Scale bar = 10 μm. **(B)** HA-mNG$_{1-10}$ cells, treated with or without doxycycline, were transfected with Sar1A-FLAG-mNG$_{11}$. The cells were fixed and stained with anti-HA and anti-FLAG antibodies. Scale bar = 10 μm.

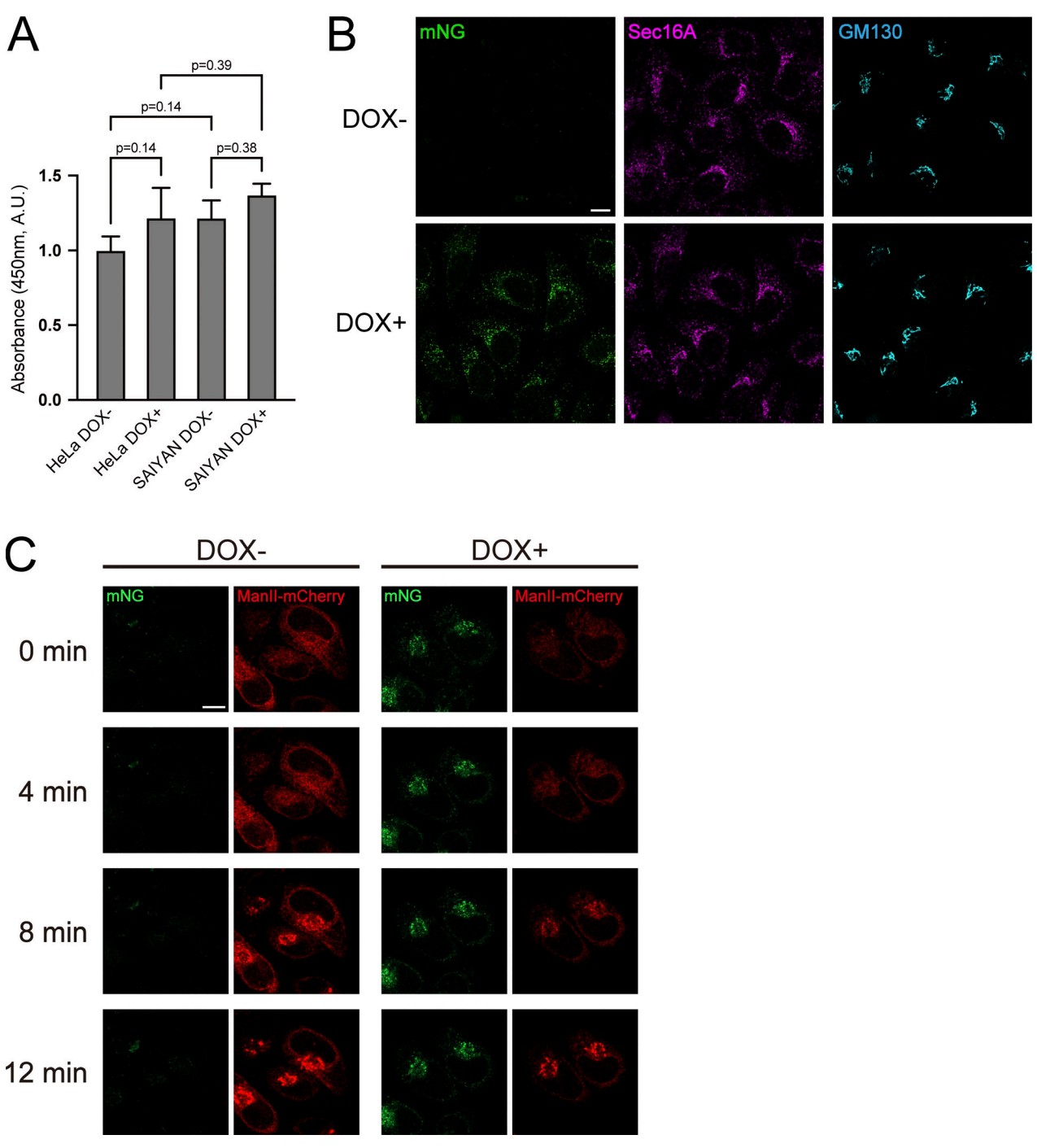

Figure S2. **Sar1A/SAIYAN (HeLa) cells proliferate and secrete normally. (A)** HeLa and Sar1A/SAIYAN (HeLa) cells were treated with or without doxycycline for 24 h, and cell viability was measured and normalized using untreated HeLa cells as control. Error bars represent the means ± SEM. $n$ = 4. **(B)** Sar1A/SAIYAN (HeLa) cells, treated with or without doxycycline, were fixed and stained with anti-Sec16-C and anti-GM130 antibodies. Scale bars = 10 μm. **(C)** Sar1A/SAIYAN (HeLa) cells treated with or without doxycycline were transfected with Str-KDEL_ManII-SBP-mCherry. RUSH chase was started with biotin addition, and live imaging was performed. Scale bars = 10 μm.

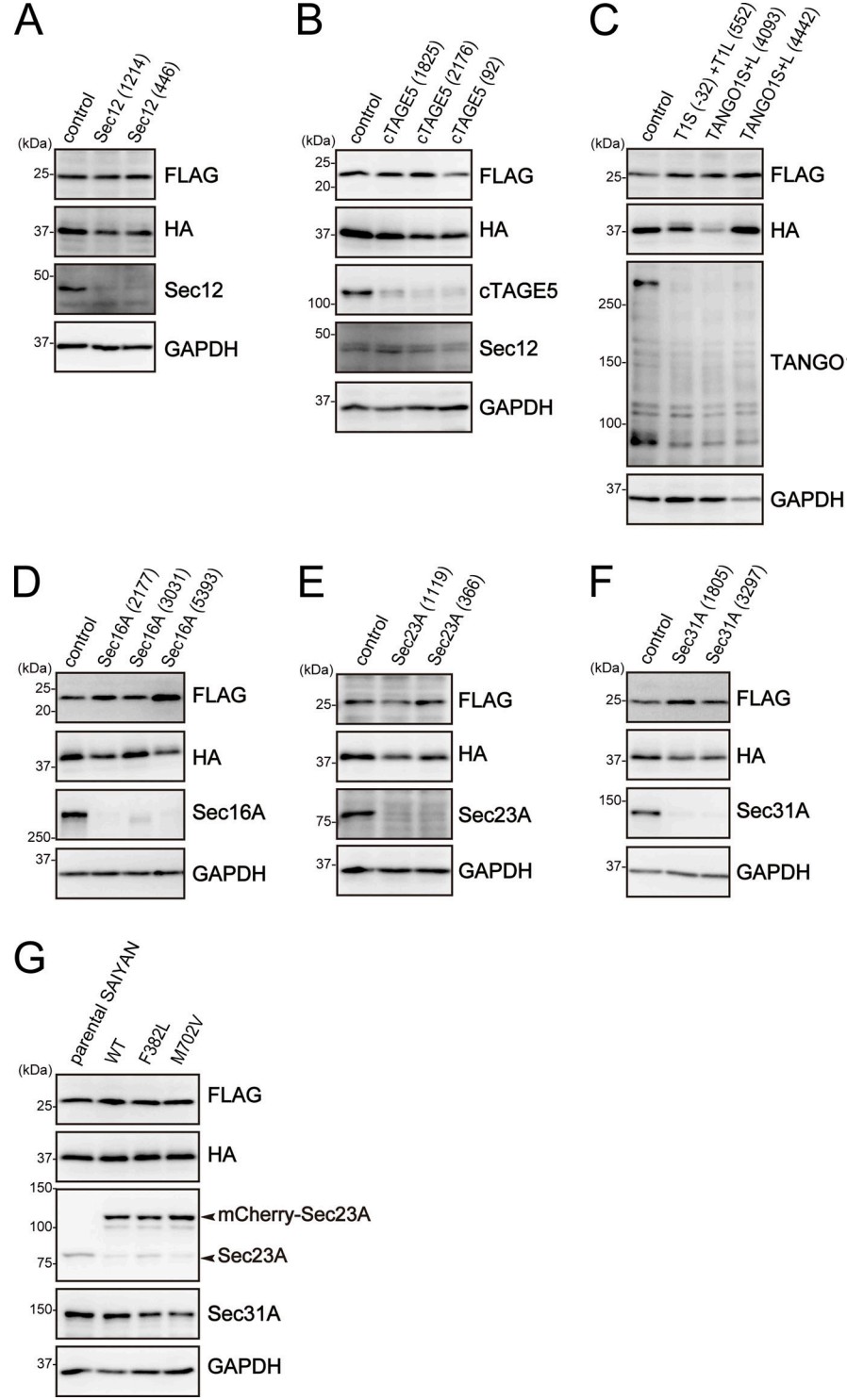

Figure S3. **Western blotting analysis of Sar1A/SAIYAN (HeLa) cells on** Figs. 3, 4, and 5. **(A)** Sar1A/SAIYAN (HeLa) cells transfected with the indicated siRNAs were lysed and subjected to SDS-PAGE, followed by western blotting with anti-FLAG, anti-HA, anti-Sec12, and anti-GAPDH antibodies. **(B)** Sar1A/ SAIYAN (HeLa) cells transfected with the indicated siRNAs were lysed and subjected to SDS-PAGE, followed by western blotting with anti-FLAG, anti-HA, anti-cTAGE5 CC1, anti-Sec12, and anti-GAPDH antibodies. **(C)** Sar1A/SAIYAN (HeLa) cells transfected with the indicated siRNAs were lysed and subjected to SDS-PAGE, followed by western blotting with anti-FLAG, anti-HA, anti-TANGO1 CC1, and anti-GAPDH antibodies. **(D)** Sar1A/SAIYAN (HeLa) cells transfected with the indicated siRNAs were lysed and subjected to SDS-PAGE, followed by western blotting with anti-FLAG, anti-HA, anti-Sec16-N, and anti-GAPDH antibodies. **(E)** Sar1A/SAIYAN (HeLa) cells transfected with the indicated siRNAs were lysed and subjected to SDS-PAGE, followed by western blotting with anti-FLAG, anti-HA, anti-Sec23A (11D8), and anti-GAPDH antibodies. **(F)** Sar1A/SAIYAN (HeLa) cells transfected with the indicated siRNAs were lysed and subjected to SDS-PAGE, followed by western blotting with anti-FLAG, anti-HA, anti-Sec31A (rabbit), and anti-GAPDH antibodies. **(G)** Sar1A/SAIYAN (HeLa) cells were stably expressed using mCherry-tagged Sec23A constructs as indicated. Cells were lysed and subjected to SDS-PAGE, followed by western blotting with anti-FLAG, anti-HA, anti-Sec23A (11D8), anti-Sec31A (rabbit), and anti-GAPDH antibodies. Source data are available for this figure: SourceData FS3.

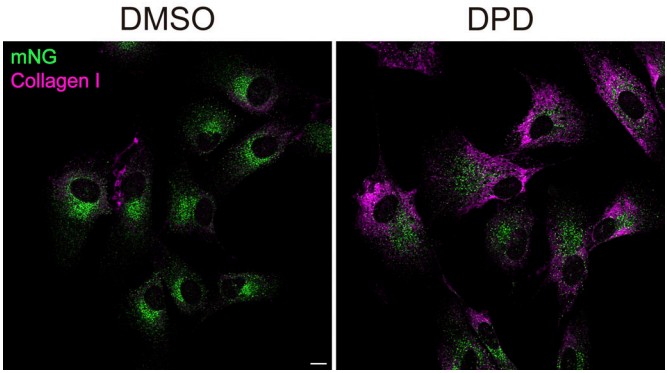

Figure S4. **DPD treatment accumulates collagen I within the ER of Sar1A/SAIYAN (BJ-5ta) cells.** Sar1A/SAIYAN (BJ-5ta) cells were treated with DMSO or 0.5 mM DPD and incubated for 16 h. Cells were fixed and stained with an anti-collagen I antibody. Scale bar = 10 µm.

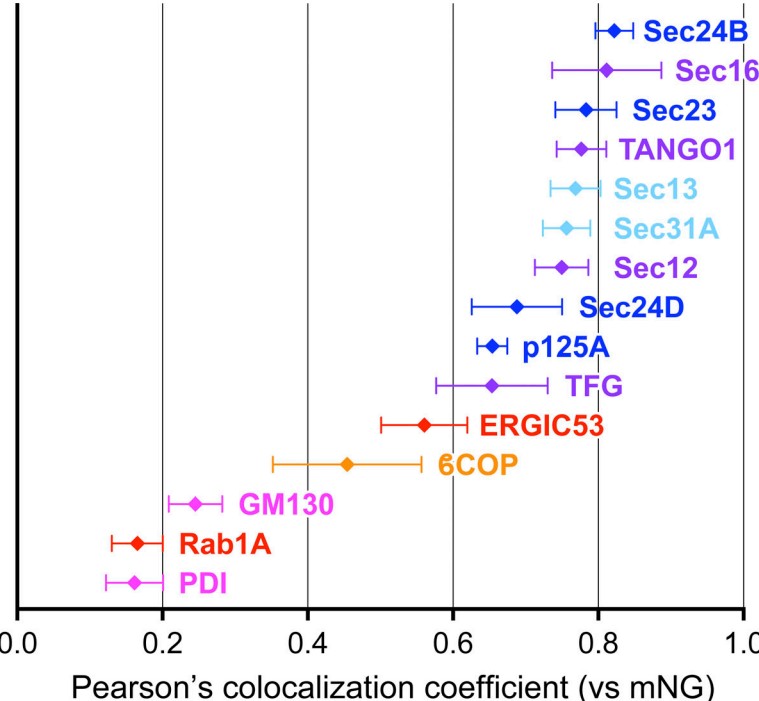

Figure S5. **Quantification of Pearson's correlation coefficient to quantify the degree of colocalization in Sar1A/SAIYAN (HeLa) cells.** Sar1A/SAIYAN (HeLa) cells were fixed and stained with anti-Sec16-C, anti-ERGIC53, anti-Sec23, anti-Sec24B, anti-Sec24D, anti-p125A, anti-TANGO1-CT, anti-Sec12, anti-TFG, anti-Sec13, anti-Sec31A (mouse), anti-β-COP, anti-GM130, anti-PDI, and anti-Rab1A antibodies. Images were captured using the Airyscan2. $n$ = 5. Cyan; outer COPII coats, blue; inner COPII coats, purple; endoplasmic reticulum (ER) exit site resident proteins, red; ERGIC proteins, orange; COPI protein, magenta; ER and Golgi proteins. Error bars represent the mean 95% CI.

**Provided online is Table S1. Table S1 shows siRNA sequences used in this study.**

