## [Peer Review File · The Journal of Cell Biology]

Small GTPase ActlvitY ANalyzing (SAIYAN) system: a method to detect GTPase activation in living cells

Miharu Maeda, Masashi Arakawa, Yukie Komatsu, and Kota Saito

Corresponding Author(s): Kota Saito, Akita University

Review Timeline:

Submission Date:	2024-03-28
Editorial Decision:	2024-04-30
Revision Received:	2024-06-10
Editorial Decision:	2024-07-02
Revision Received:	2024-07-08
Accepted:	2024-07-11

Monitoring Editor: Elizabeth Miller

Scientific Editor: Andrea Marat

Transaction Report:

DOI: <https://doi.org/10.1083/jcb.202403179>

April 30, 2024

Re: JCB manuscript #202403179

Dr. Kota Saito
Akita University
Graduate School of Medicine
1-1-1, hondo
Akita, Akita 010-8543
Japan

Dear Dr. Saito,

Thank you for submitting your manuscript entitled "New Method Visualizes Active Small GTPases, Reveals Unique Sar1 Active Sites in Collagen Secretion". The manuscript was assessed by expert reviewers, whose comments are appended to this letter. We invite you to submit a revision if you can address the reviewers' key concerns, as outlined here.

As you will see, the reviewers appreciate that your study provides a new method to examine GTPase activation in cells. However, they have concerns that require further experiments to address, specifically to confirm the functionality of the SAIYAN technique. In addition, reviewer 2 has concerns regarding your conclusions about collagen secretion. Overall, we think a suitably revised manuscript would be most appropriate as a JCB Tool as opposed to an Article. As a Tool, please moderate your conclusions regarding collagen secretion, as well as discuss the issue that interpretation of colocalization of Sec13-31 with Sar1 is complicated by the fact this may lead to enhanced GTPase activity, which is likely a specific consideration for Sar1 in the use of the SAIYAN technique. All of reviewer 1 and 3's concerns must be addressed with new experiments where requested. In particular, it is essential to demonstrate that the technique correctly recapitulates the GTPase cycle of Sar1 by providing further evidence that the modifications do not alter the GTPase activity. Specifically, you must rule out that FP complementation does not alter turnover, for example after GTP hydrolysis the reporter may have a longer half-life on the membrane due to being stabilized by the FP. This can be tested by overexpressing 13/31 to fully stimulate hydrolysis and reset the system. As noted by reviewer 3 you must also verify that membrane bound Sar1 is indeed GTP bound given that Sar1 GDP binds membranes weakly. Preferably in vitro GTPase activity assays should be used to comprehensively address these concerns.

GENERAL GUIDELINES:

Text limits: Character count for an Article is < 40,000, not including spaces. Count includes title page, abstract, introduction, results, discussion, and acknowledgments. Count does not include materials and methods, figure legends, references, tables, or supplemental legends.

Figures: Articles may have up to 10 main text figures. Figures must be prepared according to the policies outlined in our Instructions to Authors, under Data Presentation, <https://jcb.rupress.org/site/misc/ifora.xhtml>. All figures in accepted manuscripts will be screened prior to publication.

Supplemental information: There are strict limits on the allowable amount of supplemental data. Articles may have up to 5 supplemental figures. Up to 10 supplemental videos or flash animations are allowed. A summary of all supplemental material should appear at the end of the Materials and methods section.

Please note that JCB now requires authors to submit Source Data used to generate figures containing gels and Western blots with all revised manuscripts. This Source Data consists of fully uncropped and unprocessed images for each gel/blot displayed in the main and supplemental figures. Since your paper includes cropped gel and/or blot images, please be sure to provide one Source Data file for each figure that contains gels and/or blots along with your revised manuscript files. File names for Source Data figures should be alphanumeric without any spaces or special characters (i.e., SourceDataF#, where F# refers to the associated main figure number or SourceDataFS# for those associated with Supplementary figures). The lanes of the gels/blots should be labeled as they are in the associated figure, the place where cropping was applied should be marked (with a box), and molecular weight/size standards should be labeled wherever possible.

Source Data files will be made available to reviewers during evaluation of revised manuscripts and, if your paper is eventually

published in JCB, the files will be directly linked to specific figures in the published article.

The typical timeframe for revisions is three to four months. While most universities and institutes have reopened labs and allowed researchers to begin working at nearly pre-pandemic levels, we at JCB realize that the lingering effects of the COVID-19 pandemic may still be impacting some aspects of your work, including the acquisition of equipment and reagents. Therefore, if you anticipate any difficulties in meeting this aforementioned revision time limit, please contact us and we can work with you to find an appropriate time frame for resubmission. Please note that papers are generally considered through only one revision cycle, so any revised manuscript will likely be either accepted or rejected.

Thank you for this interesting contribution to Journal of Cell Biology. You can contact us at the journal office with any questions at cellbio@rockefeller.edu.

Sincerely,

Elizabeth Miller, PhD
Monitoring Editor

Andrea L. Marat, PhD
Senior Scientific Editor

Journal of Cell Biology

Reviewer #1 (Comments to the Authors (Required)):

This manuscript from the Saito lab presents a new method to probe the activation states of GTPases by using split-GFP to monitor their presence on an organelle membrane surface. By fusing the mNG11 peptide to Sar1, and expressing an ER-localized mNG1-10 construct, mNeonGreen signal can be used to detect when Sar1 is present on the ER membrane surface. As Sar1 localization is coupled to its activation, this system therefore provides a measure of Sar1 activation in cells, and also provides the localization of this activation.

The authors find that depletion of Sec12, cTAGE5, TANGO1, or Sec16A reduces the amount of activated Sar1 in cells. Based on previous studies, these results validate the probe, but they also provide new spatial information about Sar1 activation under these conditions.

The authors find that loss of Sec23A reduces the amount of activated Sar1, while loss of Sec31A increases the amount of Sar1. The finding that Sec23A knockdown results in less Sar1 activation may seem surprising, but as the authors point out, these results provide additional evidence supporting the established role of Sec31 as the critical driver of Sar1 inactivation in cells. Similarly, introduction of a Sec23A disease mutant known to disrupt interaction with Sec31 causes an increase in Sar1 activation.

Using the SAIYAN tool in collagen-secreting fibroblasts, the authors find that activated Sar1 localizes to ERGIC compartments as well as ER exit sites (ERES). They come to this conclusion based on their assessment of the morphology of the compartments as well as the co-localization pattern. They define ERGIC sites as those labeled with ERGIC-53 but not labeled with Sec16. They find that both Sec23 and activated Sar1 are present at these ERGIC compartments which are distinct from Sec16-labeled ERES.

They also investigate colocalization with other trafficking machinery, including the finding that there is some co-localization of activated Sar1 with COPI, similar to other recent reports. Notably, they find that the extent of Sar1 colocalization with different proteins varies between different cell lines.

Finally, they find that perturbing collagen folding in the ER, by chemical inhibition of proyl isomerase, results in accumulation of activated Sar1 at ERES, and loss of activated Sar1 at ERGIC compartments, which is more similar to what they observe in HeLa

cells. They interpret this to mean that the structure of the ER-Golgi interface is different in cells secreting collagen.

The authors conclude with an important discussion regarding the limitations of other recent high-profile studies reporting that cargos exit the ER in tubes or "tunnels". In this manuscript, the authors have presented data that supports some aspects of these recent studies, but provides important refinements to these models based on their new data.

Overall, I found this to be a comprehensive and carefully conducted study. I think their new tool is powerful, and has the potential to be adopted for use with other trafficking GTPases. I also think their extensive co-localization analysis using multiple cell types and when disrupting collagen folding provide important new information for the field in trying to make sense of the COPII tube/tunnel versus vesicle debate. I have only a few questions and suggestions for improvement:

1. Is the C-terminally tagged Sar1A functional? The authors state that the constructs are not toxic to cells, but Sar1B may be providing the Sar1 function when Sar1A is tagged. In the interest of rigor, it would be good if the authors were able to further address this possibility.
2. In the introduction, the authors list several existing approaches to measuring GTPase activation in cells, but they do not list FP-effector probes among them. They do discuss FP-effectors as FRET probes, but they do not mention the use of FP-effectors as simple fluorescent probes (these have also been called "relocation sensors"). For example, Rho GTPase sensors have been used by several labs. A recent publication which references and builds upon these earlier studies is: *J Cell Sci* (2021) 134 (17): jcs258823. There are probably other examples, as well.

Minor points:

3. In the abstract, the authors write that the motivation for their development of SAIYAN is that FRET is "limited to those with well-defined effectors, excluding Sar1". But Sar1 has a very well-defined effector, Sec23. The problem with Sar1, as the authors later state, is that Sec23 is also a Sar1-GAP. I think the authors should consider fixing this sentence to more accurately reflect the challenge with Sar1 that they sought to address.
4. Line 12, page 5, "bundles" should probably be a different word.

Reviewer #2 (Comments to the Authors (Required)):

Maeda et al. developed a novel method to visualize active Sar1 in cells. Using this approach, they assert that the activation of Sar1 through the ERGIC is important for collagen secretion. From a technical standpoint, the experiments were nicely conducted, incorporating appropriate controls.

My primary concern is that the observation of active Sar1 proteins in ERGIC/ERES tubules, as related to collagen secretion, is circumstantial or correlative in this study. This does not definitively prove or disprove the importance of the observation in collagen secretion.

Specific comments:

Page 5, line 28-32: Performing cell fractionation to determine whether Sar1A-FLAG-mNG11 is present in both cytosolic and membrane fractions could be beneficial here. Using endogenous Sar1 as an internal control would be advisable.

Page 10, line 3-8: I have a concern regarding the observed lower colocalization of Sec13-Sec31 with activated Sar1. The presence of Sec13-Sec31 leads to the inactivation of Sar1 due to enhanced GTP hydrolysis. Consequently, the overlap between Sec13-Sec31 and activated Sar1 is not as prominent as the overlap between Sec23-Sec24 and activated Sar1, as shown in Fig. 3c. This result suggests that a reduced signal of activated Sar1 does not necessarily indicate that Sec13-Sec31 interacts with Sar1 less than Sec23-Sec24 does. Obviously, the degree of colocalization could also be affected if certain proteins directly or indirectly influence Sar1's GTPase state. Theoretically, regions with relatively weak signals of activated Sar1 could be where Sar1 is most actively performing its function, whereas regions with stronger Sar1-GTP signals might be where Sar1-GTP is poised to act (e.g., prior to interacting with Sec13-Sec31). Therefore, equating the degree of overlap with Sar1's functionality could be misleading. I believe this represents a major issue with this method.

The term "colocalization rate" suggests a temporal measurement, which is inaccurate. It seems the authors might have intended to refer to the degree of colocalization.

Fig. 4 to Fig. 6: Essentially correlative, lacking substantial definitive information.

Fig. 6: I wonder whether the authors have considered the possibility that DPD treatment might trigger an ER stress response due to the accumulation of collagens in the ER. This induction of ER stress could potentially influence ER exit or alter the levels of ER export machinery. Perhaps this is what accounts for the observed differences in staining patterns.

Reviewer #3 (Comments to the Authors (Required)):

The manuscript by Maeda et al. describes a new technology aimed at visualizing an active GTPase by fluorescence imaging in live cells, using a split mNeonGreen (mNG) system. The authors call it the SAIYAN system. The technique monitors the level of a GTPase that is membrane-bound, which is generally the active GTP-bound form. The authors apply their new technique to Sar1, a well-studied GTPase that is essential for the formation of transport carriers at ER exit sites, the first step of the secretory pathway. However, Sar1 has been shown to bind to membranes in its GDP-bound form, albeit more weakly than in its GTP-bound form (Paul et al. 2023 PNAS 120:e2212513120). Indeed, the classic dominant negative Sar1-T39N mutant, which is unable to bind GTP, gave a positive mNG signal at the ER membrane (Fig. 1b), although the authors claim the signal is weaker than for the WT Sar1 (see detailed comments below). Hence it is not strictly correct to say that this assay monitors exclusively GTP-bound Sar1, when in fact it monitors membrane-bound Sar1. It is important to know what fraction of the signal is membrane-associated Sar1-GDP, and what fraction is active Sar1-GTP.

Using siRNA knock-downs, the authors show convincingly that the mNG fluorescence signal is dependent on expression of Sar1, is decreased when the GEF for Sar1 (Sec12) is depleted, and increased when a stimulator of Sar1 GAP activity (Sec31) is depleted. In addition, knock down of known components of ER exit sites also lead to lower Sar1 mNG signal. As a final control, the authors show that CLSD mutations in Sec23A give the expected mNG phenotype when expressed in cells. Studies in collagen-expressing cells indicated that ER exit sites have different structures depending on the cargo secreted, and demonstrate that membrane-bound active Sar1 co-localized with inner COPII components along structures linking ER exit sites and ERGIC. These results provide important information on the mechanisms by which cargo is packaged and transported through the early secretory pathway that is actually used in cells.

Detailed comments:

1. The split mNeonGreen (mNG) SAIYAN system monitors ER membrane binding of Sar1. The authors need to determine what fraction of the mNG signal is from Sar1-GTP, and what fraction from Sar1-GDP, which binds weakly to membranes, in order to conclude that the mNG signal corresponds to active GTP-bound Sar1.
2. Page 6, lines 3-5, Fig. 1b: What is localization of the Sar1-deltaN mutant? Is it cytosolic, as predicted? The image shown in Fig. 1b shows a clear difference compared to WT Sar1, but the pattern appears quite punctate.
3. Page 6, lines 7-9: The authors should explain why the Sar1-T39N mutant is localized to ER membranes. Some Arf family GTPases are soluble when they cannot bind GTP, others bind to membranes via protein-protein interactions. The authors should include as a reference Paul et al. 2023 PNAS 120:e2212513120 in describing Sar1-T39N localization to ER membranes.
4. Page 6, lines 7-11, figure 1b. The mNG signal for Sar1-T39N looks quite strong in the image provided, and given the distribution of this fluorescence throughout the ER, rather than concentrated at ER exit sites, it is not clear from the images shown that the Sar1-T39N mNG signal is really weaker than the WT mNG signal. It is also not clear from the data provided that the Sar1-H79G mNG signal is stronger than that for WT Sar1. Quantification of the Sar1-T39N mNG signal is particularly important for the conclusions drawn in this manuscript, since in fact the mNG system described here monitors membrane-bound Sar1, not just Sar1-GTP. The authors need to provide quantitative data on how much of the mNG signal is contributed by membrane-bound Sar1-GDP.
5. Page 7 line 22 and following. The authors should explain more clearly why depletion of the GAP for Sar1, Sec23, causes a decrease in Sar1-GTP levels, rather than the expected increase. There is a reference to Bi et al. 2007, but the results of this paper are not clearly explained, nor why they support the results in Fig. 3a and b. This is an important point for the paper, as the authors claim that the mNG signal corresponds to active Sar1-GTP, and Sec23 has been demonstrated to be a GAP for Sar1.
6. Page 7, lines 31-32 The authors state: "These results suggest that the SAIYAN system can detect the augmentation of small GTPases." This statement is not clear as written. I assume the authors mean that the SAIYAN system can detect an increase in the level of membrane-bound Sar1 (presumably due mostly to an increase in the active GTP-bound form) in cells, which needs to be stated explicitly.
7. The authors use the term "Sar1 activity" repeatedly (Page 5 line 10, Page 7 multiple places, page 8 line 27), but it is not clear what is meant by this term when used in these instances. Sar1 acts to recruit effectors upon activation by binding GTP. However, I assume that in most cases, the authors mean "active Sar1-GTP". Please see comments above regarding what exactly the mNG signal monitors - the most correct formulation is "membrane-bound Sar1", but with appropriate controls and quantifications, "membrane-bound active Sar1-GTP" could be justified.
8. Page 8, line 17. What do the authors mean by "intrinsic Sec13/31". Perhaps "endogenous Sec13/31"?
9. Page 10, line 2 and page 9, line 25. Do the authors really mean "dot-like" or "dotted-like" which implies perfectly spherical small shapes? I would suggest "punctate" as a better alternative.
10. Page 15, line 28-30. In the phrase "The ability to accurately measure both Sar1 inactivation and activation through knockdown of each factor served as the foremost validation...", it is not clear what factors are being referred to, and they should be indicated explicitly.

Dear Dr. Saito,

Thank you for submitting your manuscript entitled "New Method Visualizes Active Small GTPases, Reveals Unique Sar1 Active Sites in Collagen Secretion". The manuscript was assessed by expert reviewers, whose comments are appended to this letter. We invite you to submit a revision if you can address the reviewers' key concerns, as outlined here.

As you will see, the reviewers appreciate that your study provides a new method to examine GTPase activation in cells. However, they have concerns that require further experiments to address, specifically to confirm the functionality of the SAIYAN technique. In addition, reviewer 2 has concerns regarding your conclusions about collagen secretion. Overall, we think a suitably revised manuscript would be most appropriate as a JCB Tool as opposed to an Article. As a Tool, please moderate your conclusions regarding collagen secretion, as well as discuss the issue that interpretation of colocalization of Sec13-31 with Sar1 is complicated by the fact this may lead to enhanced GTPase activity, which is likely a specific consideration for Sar1 in the use of the SAIYAN technique. All of reviewer 1 and 3's concerns must be addressed with new experiments where requested. In particular, it is essential to demonstrate that the technique correctly recapitulates the GTPase cycle of Sar1 by providing further evidence that the modifications do not alter the GTPase activity. Specifically, you must rule out that FP complementation does not alter turnover, for example after GTP hydrolysis the reporter may have a longer half-life on the membrane due to being stabilized by the FP. This can be tested by overexpressing 13/31 to fully stimulate hydrolysis and reset the system. As noted by reviewer 3 you must also verify that membrane bound Sar1 is indeed GTP bound given that Sar1 GDP binds membranes weakly. Preferably in vitro GTPase activity assays should be used to comprehensively address these concerns.

GENERAL GUIDELINES:

Text limits: Character count for an Article is < 40,000, not including spaces. Count includes title page, abstract, introduction, results, discussion, and acknowledgments. Count does not include materials and methods, figure legends, references, tables, or supplemental legends.

Figures: Articles may have up to 10 main text figures. Figures must be prepared according to the policies outlined in our Instructions to Authors, under Data Presentation, <https://jcb.rupress.org/site/misc/ifora.xhtml>. All figures in accepted manuscripts will be screened prior to publication.

IMPORTANT: It is JCB policy that if requested, original data images must be made available. Failure to provide original images upon request will result in unavoidable delays in publication. Please ensure that you have access to all original microscopy and blot data images before submitting your revision.

Supplemental information: There are strict limits on the allowable amount of supplemental data. Articles may have up to 5 supplemental figures. Up to 10 supplemental videos or flash animations are allowed. A summary of all supplemental material should appear at the end of the Materials and methods section.

Please note that JCB now requires authors to submit Source Data used to generate figures containing gels and Western blots with all revised manuscripts. This Source Data consists of fully uncropped and unprocessed images for each gel/blot displayed in the main and supplemental figures. Since your paper includes cropped gel and/or blot images, please be sure to provide one Source Data file for each figure that contains gels and/or blots along with your revised manuscript files. File names for Source Data figures should be alphanumeric without any spaces or special characters (i.e., SourceDataF#, where F# refers to the associated main figure number or SourceDataFS# for those associated with Supplementary figures). The lanes of the gels/blots should be labeled as they are in the associated figure, the place where cropping was applied should be marked (with a box), and molecular weight/size standards should be labeled wherever possible.

The typical timeframe for revisions is three to four months. While most universities and institutes have reopened labs and allowed researchers to begin working at nearly pre-pandemic levels, we at JCB realize that the lingering effects of the COVID-19 pandemic may still be impacting some aspects of your work, including the acquisition of equipment and reagents. Therefore, if you anticipate any difficulties in meeting this aforementioned revision time limit, please contact us and we can work with you to find an appropriate time frame for resubmission. Please note that papers are generally considered through only one revision cycle, so any revised manuscript will likely be either accepted or rejected.

When submitting the revision, please include a cover letter addressing the reviewers' comments point by

point. Please also highlight all changes in the text of the manuscript.

Thank you for this interesting contribution to Journal of Cell Biology. You can contact us at the journal office with any questions at cellbio@rockefeller.edu.

Sincerely,

Elizabeth Miller, PhD
Monitoring Editor

Andrea L. Marat, PhD
Senior Scientific Editor

Journal of Cell Biology

We thank you for the valuable insights that have been provided by the reviewers and editors. As per your suggestion, we have changed the manuscript from a JCB article to fit the requirements of JCB tools and toned down the conclusions regarding collagen secretion. Consequently, we have removed references to collagen secretion from the title. Additionally, we have toned down the statements about collagen secretion in both the abstract and the discussion.

In response to Reviewer #2's comments, we have addressed the potential reasons for the low degree of colocalization between Sec13/31 and active Sar1. Specifically, we discussed the possibility that Sec13/31's GAP-enhancing activity leads to Sar1 inactivation, which is now included in the Discussion section on Lines 1–5 on page 15.

To demonstrate that SAIYAN does not affect Sar1 turnover, we investigated the status of active Sar1 upon Sec13/Sec31 overexpression, as you suggested. As shown in Fig. 4E and 4F, Sec13/Sec31 overexpression resulted in the mNG signal decreasing to levels comparable to Sar1 T39N expression and Sec12 KD. This finding indicates that SAIYAN can detect signal reduction due to inactivation, even after Sar1 has been activated and bound to the membrane, forming the complete mNG.

Furthermore, as pointed out by Reviewer #3, we agree that the SAIYAN system measures the membrane-

bound state of Sar1 rather than its nucleotide form. Nonetheless, as GDP-bound Sar1 binds to membranes more weakly than GTP-bound Sar1, our validation of the SAIYAN system showed that the T39N Sar1 mutant exhibited approximately half the mNG signal compared to WT Sar1 (Fig. 1C). This result is consistent with the mNG signal levels observed when Sec12 is knocked down (Fig. 3B) or when Sec13/Sec31 (Fig. 4F) is overexpressed. Thus, while the SAIYAN system measures the amount of membrane-bound Sar1, it can still detect shifts in Sar1's nucleotide form based on whether the mNG signal increases or decreases following a particular treatment.

Quantitative measurement of nucleotide forms requires the use of other methods, such as metabolic labeling, in conjunction with the SAIYAN system. Each method has its advantages and disadvantages. Metabolic labeling involves biochemically disrupting cells to measure nucleotides, which necessitates consideration of GTPase hydrolysis in the lysate. The SAIYAN system, in contrast, offers the advantage of being simple and capable of detecting Sar1 activation sites in live cells, though it lacks strict quantitative accuracy.

In addition, SAIYAN offers an advantage over existing relocation biosensors, where any fluorescence in the cytoplasm that is not bound to the activated GTPase is considered background. In contrast, SAIYAN is expected to reduce background fluorescence because it only emits fluorescence when the small GTPase is membrane-bound.

In any case, we believe that the SAIYAN system holds value as a novel method for measuring activation status of GTPases.

Reviewer #1 (Comments to the Authors (Required)):

This manuscript from the Saito lab presents a new method to probe the activation states of GTPases by using split-GFP to monitor their presence on an organelle membrane surface. By fusing the mNG11 peptide to Sar1, and expressing an ER-localized mNG1-10 construct, mNeonGreen signal can be used to detect when Sar1 is present on the ER membrane surface. As Sar1 localization is coupled to its activation, this system therefore provides a measure of Sar1 activation in cells, and also provides the localization of this activation.

The authors find that depletion of Sec12, cTAGE5, TANGO1, or Sec16A reduces the amount of activated Sar1 in cells. Based on previous studies, these results validate the probe, but they also provide new spatial information about Sar1 activation under these conditions.

The authors find that loss of Sec23A reduces the amount of activated Sar1, while loss of Sec31A increases the amount of Sar1. The finding that Sec23A knockdown results in less Sar1 activation may seem surprising, but as the authors point out, these results provide additional evidence supporting the established role of Sec31 as the critical driver of Sar1 inactivation in cells. Similarly, introduction of a Sec23A disease mutant known to disrupt interaction with Sec31 causes an increase in Sar1 activation.

Using the SAIYAN tool in collagen-secreting fibroblasts, the authors find that activated Sar1 localizes to ERGIC compartments as well as ER exit sites (ERES). They come to this conclusion based on their assessment of the morphology of the compartments as well as the co-localization pattern. They define ERGIC sites as those labeled with ERGIC-53 but not labeled with Sec16. They find that both Sec23 and activated Sar1 are present at these ERGIC compartments which are distinct from Sec16-labeled ERES.

They also investigate colocalization with other trafficking machinery, including the finding that there is some co-localization of activated Sar1 with COPI, similar to other recent reports. Notably, they find that the extent of Sar1 colocalization with different proteins varies between different cell lines.

Finally, they find that perturbing collagen folding in the ER, by chemical inhibition of prolyl isomerase, results in accumulation of activated Sar1 at ERES, and loss of activated Sar1 at ERGIC compartments, which is more similar to what they observe in HeLa cells. They interpret this to mean that the structure of the ER-Golgi interface is different in cells secreting collagen.

The authors conclude with an important discussion regarding the limitations of other recent high-profile studies reporting that cargos exit the ER in tubes or "tunnels". In this manuscript, the authors have presented data that supports some aspects of these recent studies, but provides important refinements to these models based on their new data.

Overall, I found this to be a comprehensive and carefully conducted study. I think their new tool is powerful, and has the potential to be adopted for use with other trafficking GTPases. I also think their extensive co-localization analysis using multiple cell types and when disrupting collagen folding provide important new information for the field in trying to make sense of the COPII tube/tunnel versus vesicle debate. I have only a few questions and suggestions for improvement:

Thank you very much for your positive and supportive review comments. We are also very grateful for the instances in which you have pointed out the shortcomings in the explanation of existing systems for detecting GTPase activation. Your feedback has helped us make the manuscript more comprehensive and accurate. As noted below, we have made several changes in accordance with your advice.

1. Is the C-terminally tagged Sar1A functional? The authors state that the constructs are not toxic to cells, but Sar1B may be providing the Sar1 function when Sar1A is tagged. In the interest of rigor, it would be good if the authors were able to further address this possibility.

Thank you for this comment. To further confirm that Sar1A/SAIYAN (HeLa) cells do not exhibit toxicity due to system detection, we examined the secretion from the ER to the Golgi apparatus using the RUSH system with and without the addition of doxycycline. As shown in Fig. S2C, the transport of ManII from the ER to the Golgi apparatus is unchanged with or without doxycycline, and this secretion rate is also consistent with that of the parental HeLa cells used. Therefore, the environment in which the mNG signal can be detected in Sar1A/SAIYAN (HeLa) cells does not show any particular toxicity.

Furthermore, to investigate the effect of Sar1B as pointed out, we conducted Sar1A knockdown in both parental HeLa and Sar1A/SAIYAN (HeLa) cells and examined the expression levels of Sar1B during this process. The results showed that, regardless of the presence or absence of doxycycline and the knockdown of Sar1A, there was no significant difference in the expression levels of Sar1B. This suggests that it is unlikely that Sar1B compensates for the function of Sar1A tagged at the C-terminus. Additionally, there was no increase in the expression levels of intact Sar1A in Sar1A/SAIYAN cells due to the partial C-terminal tagging of Sar1A, indicating that Sar1A with a C-terminal tag is functional.

2. In the introduction, the authors list several existing approaches to measuring GTPase activation in cells, but they do not list FP-effector probes among them. They do discuss FP-effectors as FRET probes, but they do not mention the use of FP-effectors as simple fluorescent probes (these have also been called "relocation sensors"). For example, Rho GTPase sensors have been used by several labs. A recent publication which references and builds upon these earlier studies is: J Cell Sci (2021) 134 (17): jcs258823. There are probably other examples, as well.

Thank you very much for your insightful comment. We have added the reference you have provided regarding the FP-effector probe and included additional details in the introduction. Please refer to lines 2–25 on page 4. This has allowed us to provide a more in-depth description of the system for detecting GTPase activation.

Minor points:

3. In the abstract, the authors write that the motivation for their development of SAIYAN is that FRET is "limited to those with well-defined effectors, excluding Sar1". But Sar1 has a very well-defined effector, Sec23. The problem with Sar1, as the authors later state, is that Sec23 is also a Sar1-GAP. I think the authors should consider fixing this sentence to more accurately reflect the challenge with Sar1 that they sought to address.

Thank you for your feedback on this. Indeed, we agree that Sec23 is a well-established effector of Sar1. Therefore, we have made corrections to the Abstract and all relevant sections accordingly.

4. Line 12, page 5, "bundles" should probably be a different word.

Thank you for pointing out. We have changed "bundles" to "strands."

Reviewer #2 (Comments to the Authors (Required)):

Maeda et al. developed a novel method to visualize active Sar1 in cells. Using this approach, they assert that the activation of Sar1 through the ERGIC is important for collagen secretion. From a technical standpoint, the experiments were nicely conducted, incorporating appropriate controls.

My primary concern is that the observation of active Sar1 proteins in ERGIC/ERES tubules, as related to collagen secretion, is circumstantial or correlative in this study. This does not definitively prove or disprove the importance of the observation in collagen secretion.

I greatly appreciate the acknowledgment of the appropriate experimental controls and their execution. Your

insightful point regarding the description of collagen secretion, which merely suggests correlation rather than definitively proving the importance of activated Sar1 in the ERGIC, is greatly appreciated. We agree that our results demonstrate the presence of activated Sar1 in the ERGIC region, leaving the elucidation of its specific functions for future investigation.

Following the recommendation of the editors, we have changed the manuscripts to JCB tools and have also updated the title of the paper to "A novel method to detect the activation of small GTPases in living cells." We will introduce the activation state of Sar1 in collagen secretion as a specific example verifiable through the SAIYAN system. We believe that our findings, while descriptive in nature, are still worth reporting when considering recent insights into the secretion process from ER exit sites to the Golgi apparatus. We will now address each specific comment provided and respond accordingly.

Specific comments:

Page 5, line 28-32: Performing cell fractionation to determine whether Sar1A-FLAG-mNG₁₁ is present in both cytosolic and membrane fractions could be beneficial here. Using endogenous Sar1 as an internal control would be advisable.

Thank you very much for your excellent suggestion. We conducted the proposed experiment and presented the results in Fig. 2G. Both intact Sar1A and Sar1A-FLAG-mNG₁₁ were mostly located in the cytoplasmic fraction, but some were observed to be localized in the membrane fraction. Importantly, even when doxycycline was added to induce the expression of HA-mNG₁₋₁₀, there was no change in the ratio of Sar1A-FLAG-mNG₁₁ between the cytoplasm and membrane fractions. This result strongly suggests that the Sar1A-FLAG-mNG₁₁ present in the cytoplasm is not artificially recruited to the membrane to form the complete mNG with HA-mNG₁₋₁₀.

Page 10, line 3-8: I have a concern regarding the observed lower colocalization of Sec13-Sec31 with activated Sar1. The presence of Sec13-Sec31 leads to the inactivation of Sar1 due to enhanced GTP hydrolysis. Consequently, the overlap between Sec13-Sec31 and activated Sar1 is not as prominent as the overlap between Sec23-Sec24 and activated Sar1, as shown in Fig. 3c. This result suggests that a reduced signal of activated Sar1 does not necessarily indicate that Sec13-Sec31 interacts with Sar1 less than Sec23-Sec24 does. Obviously, the degree of colocalization could also be affected if certain proteins directly or indirectly influence Sar1's GTPase state. Theoretically, regions with relatively weak signals of activated Sar1 could be where Sar1 is most actively performing its function, whereas regions with stronger Sar1-GTP signals might be where Sar1-GTP is poised to act (e.g., prior to interacting with Sec13-Sec31). Therefore, equating the degree of overlap with Sar1's functionality could be misleading. I believe this represents a major issue with this method.

Thank you for your suggestion. We have discussed this matter in lines 1–5 on page 15. Additionally, we have toned down the description regarding collagen secretion (lines 13–15 in the Abstract). Furthermore, we have revised the manuscript to be submitted to JCB Tools instead of JCB Articles. The main focus of this paper is now the proposal of a new method called SAIYAN technology, with the application example showing the activation site of Sar1 during collagen secretion.

The term "colocalization rate" suggests a temporal measurement, which is inaccurate. It seems the authors might have intended to refer to the degree of colocalization.

Thank you for pointing this out. Accordingly, we have changed "colocalization rate" to "the degree of colocalization" to better describe our intended meaning.

Fig. 4 to Fig. 6: Essentially correlative, lacking substantial definitive information.

Thank you for your insightful comments. We agree that these results are correlative and not definitive. However, given that this result is one application of the SAIYAN system and considering the high interest in ER-Golgi transport carriers, we believe that it is useful to report these findings. Therefore, we have removed the mention of collagen secretion from the title and toned down the mention thereof in the discussion. Additionally, following the editor's advice, we will change the manuscript type from a JCB article to JCB tools.

Fig. 6: I wonder whether the authors have considered the possibility that DPD treatment might trigger an ER stress response due to the accumulation of collagens in the ER. This induction of ER stress could potentially influence ER exit or alter the levels of ER export machinery. Perhaps this is what accounts for the observed differences in staining patterns.

Thank you very much for your excellent suggestion. To investigate the possibility that the phenotype of Sar1 during DPD treatment depends on ER stress, we treated the cells with tunicamycin, a commonly used ER stress inducer, to see if a similar phenotype would appear. As shown in the figure below, Sar1 remained in the ERGIC region after tunicamycin treatment and did not disappear from the ERGIC region as it did during DPD treatment. Therefore, it seems likely that the phenotype caused by DPD treatment is due to the inhibition of collagen secretion and may not be due to general ER stress.

A**B**
Reticular pattern of activated Sar1 signals is unchanged with tunicamycin treatment in Sar1A/SAIYAN (BJ-5ta) cells.

(A) Sar1A/SAIYAN (BJ-5ta) cells were treated with DMSO or 0.5 mM DPD or 100 ng/ml tunicamycin and incubated for 16 h. Cells were fixed and stained with anti-Sec16-C antibodies. Images were captured using the Airyscan2. Scale bars = 10 μ m. (B) Sar1A/SAIYAN (BJ-5ta) cells treated with DMSO or 0.5 mM DPD or 100 μ g/ml tunicamycin and incubated for 16 h were extracted and subjected to SDS-PAGE, followed by western blotting with anti-Bip and anti-GAPDH antibodies.

Reviewer #3 (Comments to the Authors (Required)):

The manuscript by Maeda et al. describes a new technology aimed at visualizing an active GTPase by fluorescence imaging in live cells, using a split mNeonGreen (mNG) system. The authors call it the SAIYAN system. The technique monitors the level of a GTPase that is membrane-bound, which is generally the active GTP-bound form. The authors apply their new technique to Sar1, a well-studied GTPase that is essential for the formation of transport carriers at ER exit sites, the first step of the secretory pathway. However, Sar1 has been shown to bind to membranes in its GDP-bound form, albeit more weakly than in its GTP-bound form (Paul et al. 2023 PNAS 120:e2212513120). Indeed, the classic dominant negative Sar1-T39N mutant, which is unable to bind GTP, gave a positive mNG signal at the ER membrane (Fig. 1b), although the authors claim the signal is weaker than for the WT Sar1 (see detailed comments below). Hence it is not strictly correct to say that this assay monitors exclusively GTP-bound Sar1, when in fact it monitors membrane-bound Sar1. It is important to know what fraction of the signal is membrane-associated Sar1-GDP, and what fraction is active Sar1-GTP.

Using siRNA knock-downs, the authors show convincingly that the mNG fluorescence signal is dependent on expression of Sar1, is decreased when the GEF for Sar1 (Sec12) is depleted, and increased when a stimulator of Sar1 GAP activity (Sec31) is depleted. In addition, knock down of known components of ER exit sites also lead to lower Sar1 mNG signal. As a final control, the authors show that CLSD mutations in Sec23A give the expected mNG phenotype when expressed in cells. Studies in collagen-expressing cells indicated that ER exit sites have different structures depending on the cargo secreted, and demonstrate that membrane-bound active Sar1 co-localized with inner COPII components along structures linking ER exit sites and ERGIC. These results provide important information on the mechanisms by which cargo is packaged and transported through the early secretory pathway that is actually used in cells.

Thank you very much for pointing out the knowledge that Sar1 can bind to membranes even in its GDP-bound form. In line with your suggestion, we have revised the content of the paper to clarify what the SAIYAN system detects. We entirely agree with your opinion that the SAIYAN system detects the amount of membrane-bound Sar1.

As detailed in our response to your specific comments, although Sar1 binds to membranes in its GDP-bound form, the binding is stronger in its GTP-bound form. Therefore, we believe the SAIYAN system can detect changes in the nucleotide form of Sar1. Additionally, it is effective at detecting activation sites. Despite the presence of background signals from the GDP-bound form, the SAIYAN system is valuable for its ability to measure changes conveniently and timeously in the activation state of GTPases within cells. However, we acknowledge that it cannot quantitatively measure the nucleotide form within cells. Your comments have

significantly enhanced the accuracy and completeness of the paper, and we are truly grateful to you for the same.

Detailed comments:

1. The split mNeonGreen (mNG) SAIYAN system monitors ER membrane binding of Sar1. The authors need to determine what fraction of the mNG signal is from Sar1-GTP, and what fraction from Sar1-GDP, which binds weakly to membranes, in order to conclude that the mNG signal corresponds to active GTP-bound Sar1.

We agree that the SAIYAN system measures the membrane-bound state of Sar1 rather than its nucleotide form. Nonetheless, as GDP-bound Sar1 binds to membranes more weakly than GTP-bound Sar1, our validation of the SAIYAN system showed that the T39N Sar1 mutant exhibited approximately half the mNG signal compared to WT Sar1 (Fig 1C). This result is consistent with the mNG signal levels observed when Sec12 is knocked down (Fig. 3B) or when Sec13/Sec31 (Fig. 4F) is overexpressed. Thus, while the SAIYAN system measures the amount of membrane-bound Sar1, it can still detect shifts in Sar1's nucleotide form based on whether the mNG signal increases or decreases following a particular treatment.

Quantitative measurement of nucleotide forms requires the use of other methods, such as metabolic labeling, in conjunction with the SAIYAN system. Each method has its advantages and disadvantages. Metabolic labeling involves biochemically disrupting cells to measure nucleotides, which necessitates consideration of GTPase hydrolysis in the lysate. The SAIYAN system, in contrast, offers the advantage of being simple and capable of detecting Sar1 activation sites in live cells, though it lacks strict quantitative accuracy.

In addition, SAIYAN offers an advantage over existing relocation biosensors, where any fluorescence in the cytoplasm that is not bound to the activated GTPase is considered background. In contrast, SAIYAN is expected to reduce background fluorescence because it only emits fluorescence when the small GTPase is membrane-bound.

In any case, we believe that the SAIYAN system holds value as a novel method for measuring the activation status of GTPases.

2. Page 6, lines 3-5, Fig. 1b: What is localization of the Sar1-deltaN mutant? Is it cytosolic, as predicted? The image shown in Fig. 1b shows a clear difference compared to WT Sar1, but the pattern appears quite punctate.

We believe that the Sar1-deltaN-FLAG staining image shown in Fig. 1B indicates the localization of a protein in the cytoplasm. To confirm this, we examined whether the deltaN-Sar1-FLAG-mNG₁₁ construct was

localized in the cytosolic fraction or the membrane fraction under conditions where mNG₁₋₁₀ was expressed on the ER membrane through fractionation. The figure shown below reveals that while WT Sar1 was present in small amounts in the membrane fraction, deltaN Sar1 did not migrate to the membrane fraction at all.

3. Page 6, lines 7-9: The authors should explain why the Sar1-T39N mutant is localized to ER membranes. Some Arf family GTPases are soluble when they cannot bind GTP, others bind to membranes via protein-protein interactions. The authors should include as a reference Paul et al. 2023 PNAS 120:e2212513120 in describing Sar1-T39N localization to ER membranes.

Thank you very much for your insightful comments. We have added the paper you have suggested and clarified that Sar1-T39N is likely in a GDP-bound state with weak binding to the ER membrane. Please refer to page 6, lines 3–21.

4. Page 6, lines 7-11, figure 1b. The mNG signal for Sar1-T39N looks quite strong in the image provided, and given the distribution of this fluorescence throughout the ER, rather than concentrated at ER exit sites, it is not clear from the images shown that the Sar1-T39N mNG signal is really weaker than the WT mNG signal. It is also not clear from the data provided that the Sar1-H79G mNG signal is stronger than that for

WT Sar1. Quantification of the Sar1-T39N mNG signal is particularly important for the conclusions drawn in this manuscript, since in fact the mNG system described here monitors membrane-bound Sar1, not just Sar1-GTP. The authors need to provide quantitative data on how much of the mNG signal is contributed by membrane-bound Sar1-GDP.

Thank you very much for your insightful comments. We have quantified the data presented in Fig. 1B and included the results as Fig. 1C. Additionally, we have incorporated a discussion of the quantification results for the Sar1 mutants in the main text, clarifying that the mNG signal reflects the amount of membrane-bound Sar1. Please refer to page 6, lines 3–21.

The SAIYAN system cannot precisely quantify the nucleotide forms of Sar1, but it does allow us to discuss changes in the proportion of GTP-bound or GDP-bound forms based on mNG signal variations.

While this system alone does not completely elucidate the dynamics of Sar1 GTPase, it offers a novel and straightforward method to monitor the changes in nucleotide forms and activation sites of GTPase in real-time, which was not achievable with previous methods. Thus, we believe the SAIYAN system is valuable for its unique capabilities.

5. Page 7 line 22 and following. The authors should explain more clearly why depletion of the GAP for Sar1, Sec23, causes a decrease in Sar1-GTP levels, rather than the expected increase. There is a reference to Bi et al. 2007, but the results of this paper are not clearly explained, nor why they support the results in Fig. 3a and b. This is an important point for the paper, as the authors claim that the mNG signal corresponds to active Sar1-GTP, and Sec23 has been demonstrated to be a GAP for Sar1.

Thank you for your suggestion. We have provided a detailed explanation of this section in the Discussion. Please refer to pages 13, lines 17–31.

“Sec23 has been considered a dual-function protein because it exhibits GAP activity towards Sar1 *in vitro* (Yoshihisa et al., 1993) and, together with Sec24, binds activated Sar1 and cargo receptors bound to Sec24 to form a pre-budding complex (Bi et al., 2002; Kuehn et al., 1998; Sato, 2004). Additionally, Sec13/Sec31 enhances the GAP activity of Sec23 towards Sar1 by about 10-fold (Bi et al., 2007). To date, the implications of these findings in mammalian cells have not been clearly elucidated. In this study, we observed that in Sec23 knockdown cells, activated Sar1 was downregulated. Conversely, the knockdown of Sec13/Sec31 upregulated the activated Sar1. These results suggest that Sec23/Sec24 alone do not function as GAPs within cells but rather play a role in the formation of the pre-budding complex. Only in the presence of Sec13/Sec31 does Sec23/Sec24 undergo a conformational change,

allowing them to exhibit sufficient GAP activity for Sar1 hydrolysis (Bi et al., 2007). Moreover, the decrease in the mNG signal detected by SAIYAN upon Sec13/Sec31 overexpression indicates that the Sar1-FLAG-mNG₁₁ and HA-mNG₁₋₁₀ complex formation is not irreversible and can be disrupted by Sec13/Sec31..”

6. Page 7, lines 31-32 The authors state: "These results suggest that the SAIYAN system can detect the augmentation of small GTPases." This statement is not clear as written. I assume the authors mean that the SAIYAN system can detect an increase in the level of membrane-bound Sar1 (presumably due mostly to an increase in the active GTP-bound form) in cells, which needs to be stated explicitly.

Thank you for your suggestion. Accordingly, we have removed the expression.

7. The authors use the term "Sar1 activity" repeatedly (Page 5 line 10, Page 7 multiple places, page 8 line 27), but it is not clear what is meant by this term when used in these instances. Sar1 acts to recruit effectors upon activation by binding GTP. However, I assume that in most cases, the authors mean "active Sar1-GTP". Please see comments above regarding what exactly the mNG signal monitors - the most correct formulation is "membrane-bound Sar1", but with appropriate controls and quantifications, "membrane-bound active Sar1-GTP" could be justified.

Thank you for your feedback. We have revised the ambiguous expression "Sar1 activity" according to each context. As you had noted, we also agree that the mNG signal directly assesses the amount of membrane-bound Sar1. However, I would appreciate it if you could agree that although qualitative, insights into the activation state of Sar1 can be obtained by comparing changes in this quantity.

8. Page 8, line 17. What do the authors mean by "intrinsic Sec13/31". Perhaps "endogenous Sec13/31"?

Thank you for pointing this out. We have changed the phrasing from "intrinsic Sec13/31" to "endogenous Sec13/31" accordingly.

9. Page 10, line 2 and page 9, line 25. Do the authors really mean "dot-like" or "dotted-like" which implies perfectly spherical small shapes? I would suggest "punctate" as a better alternative.

Thank you for pointing this out. We have changed the phrasing from "dot-like, dotted-like" to "punctate" accordingly.

10. Page 15, line 28-30. In the phrase "The ability to accurately measure both Sar1 inactivation and activation through knockdown of each factor served as the foremost validation...", it is not clear what factors are being

referred to, and they should be indicated explicitly.

Thank you for pointing out. We have revised the text extensively to eliminate this expression.

July 2, 2024

RE: JCB Manuscript #202403179R

Dr. Kota Saito
Akita University
Graduate School of Medicine
1-1-1, hondo
Akita, Akita 010-8543
Japan

Dear Dr. Saito:

Thank you for submitting your revised manuscript entitled "A novel method to detect the activation of small GTPases in living cells". We would be happy to publish your paper in JCB pending final revisions necessary to meet our formatting guidelines (see details below).

A. MANUSCRIPT ORGANIZATION AND FORMATTING:

1) Text limits: Character count for Tools is < 40,000, not including spaces. Count includes abstract, introduction, results, discussion, and acknowledgments. Count does not include title page, figure legends, materials and methods, references, tables, or supplemental legends.

2) Figures limits: Tools may have up to 10 main text figures.

3) Figure formatting: * Scale bars must be present on all microscopy images, including inset magnifications (you may alternatively indicate the diameter of the inset) *. Molecular weight or nucleic acid size markers must be included on all gel electrophoresis.

4) Statistical analysis: Error bars on graphic representations of numerical data must be clearly described in the figure legend. The number of independent data points (n) represented in a graph must be indicated in the legend. Statistical methods should be explained in full in the materials and methods. For figures presenting pooled data the statistical measure should be defined in the figure legends. Please also be sure to indicate the statistical tests used in each of your experiments (either in the figure legend itself or in a separate methods section) as well as the parameters of the test (for example, if you ran a t-test, please indicate if it was one- or two-sided, etc.). Also, if you used parametric tests, please indicate if the data distribution was tested for normality (and if so, how). If not, you must state something to the effect that "Data distribution was assumed to be normal but this was not formally tested."

5) Abstract and title: The abstract should be no longer than 160 words and should communicate the significance of the paper for a general audience. The title should be less than 100 characters including spaces. Make the title concise but accessible to a general readership.

* The use of "novel" is not permitted in titles. It would also be useful to include the name of the technique in the title, for example: Small GTPase Activity Analyzing (SAIYAN) system: a method to detect GTPase activation in living cells

6) Materials and methods: Should be comprehensive and not simply reference a previous publication for details on how an experiment was performed. Please provide full descriptions in the text for readers who may not have access to referenced manuscripts.

7) All antibodies, cell lines, animals, and tools used in the manuscript should be described in full, including accession numbers for materials available in a public repository such as the Resource Identification Portal. Please be sure to provide the sequences for all of your primers/oligos and RNAi constructs in the materials and methods. You must also indicate in the methods the source, species, and catalog numbers (where appropriate) for all of your antibodies. Please also indicate the acquisition and quantification methods for immunoblotting/western blots.

8) Microscope image acquisition: The following information must be provided about the acquisition and processing of images:

- a. Make and model of microscope
- b. Type, magnification, and numerical aperture of the objective lenses
- c. Temperature
- d. Imaging medium
- e. Fluorochromes
- f. Camera make and model
- g. Acquisition software
- h. Any software used for image processing subsequent to data acquisition. Please include details and types of operations involved (e.g., type of deconvolution, 3D reconstitutions, surface or volume rendering, gamma adjustments, etc.).

10) Supplemental materials: There are strict limits on the allowable amount of supplemental data. Tools may have up to 5 supplemental figures. Please also note that tables, like figures, should be provided as individual, editable files. A summary of all supplemental material should appear at the end of the Materials and methods section.

13) ORCID IDs: ORCID IDs are unique identifiers allowing researchers to create a record of their various scholarly contributions in a single place. Please note that ORCID IDs are now *required* for all authors. At resubmission of your final files, please be sure to provide your ORCID ID and those of all co-authors.

Please note that JCB now requires authors to submit Source Data used to generate figures containing gels and Western blots with all revised manuscripts. This Source Data consists of fully uncropped and unprocessed images for each gel/blot displayed in the main and supplemental figures. Since your paper includes cropped gel and/or blot images, please be sure to provide one Source Data file for each figure that contains gels and/or blots along with your revised manuscript files. File names for Source Data figures should be alphanumeric without any spaces or special characters (i.e., SourceDataF#, where F# refers to the associated main figure number or SourceDataFS# for those associated with Supplementary figures). The lanes of the gels/blots should be labeled as they are in the associated figure, the place where cropping was applied should be marked (with a box), and molecular weight/size standards should be labeled wherever possible.

Journal of Cell Biology now requires a data availability statement for all research article submissions. These statements will be published in the article directly above the Acknowledgments. The statement should address all data underlying the research presented in the manuscript. Please visit the JCB instructions for authors for guidelines and examples of statements at (<https://rupress.org/jcb/pages/editorial-policies#data-availability-statement>).

B. FINAL FILES:

-- Cover images: If you have any striking images related to this story, we would be happy to consider them for inclusion on the

journal cover. Submitted images may also be chosen for highlighting on the journal table of contents or JCB homepage carousel. Images should be uploaded as TIFF or EPS files and must be at least 300 dpi resolution.

****It is JCB policy that if requested, original data images must be made available to the editors. Failure to provide original images upon request will result in unavoidable delays in publication. Please ensure that you have access to all original data images prior to final submission.****

****The license to publish form must be signed before your manuscript can be sent to production. A link to the electronic license to publish form will be sent to the corresponding author only. Please take a moment to check your funder requirements before choosing the appropriate license.****

Thank you for your attention to these final processing requirements. Please revise and format the manuscript and upload materials within 7 days. If you need an extension for whatever reason, please let us know and we can work with you to determine a suitable revision period.

Thank you for this interesting contribution, we look forward to publishing your paper in Journal of Cell Biology.

Sincerely,

Elizabeth Miller, PhD
Monitoring Editor

Andrea L. Marat, PhD
Senior Scientific Editor

Journal of Cell Biology

Reviewer #1 (Comments to the Authors (Required)):

I think the authors have further improved the manuscript with their revisions. Overall I think this is an interesting paper reporting a clever and useful new method, as well as some interesting new biology generated with the tool. I recommend publication in JCB.

Reviewer #3 (Comments to the Authors (Required)):

The authors have addressed all the points that I raised in an excellent and comprehensive manner, and the manuscript is much improved. I have no further comments.